# Multi-Objective Golden Flower Optimization Algorithm for Sustainable Reconfiguration of Power Distribution Network with Decentralized Generation

**Dhivya Swaminathan *** and **Arul Rajagopalan ***

School of Electrical Engineering, Vellore Institute of Technology, Chennai 600127, Tamil Nadu, India
* Correspondence: dhivya.s2019@vitstudent.ac.in (D.S.); arulphd@yahoo.co.in (A.R.)

**Abstract:** This paper provides a meta-heuristic hybridized version called multi-objective golden flower pollination algorithm (MOGFPA) as the best method for choosing the optimal reconfiguration for distribution networks (DNs) in order to reduce power losses (PLs). Aside from PLs, another parameter is considered: the load balance index (LBI). The expression for the LBI is stated using real and reactive indices. It makes the optimal distributed generation (DG) placement and DN routing of the multi-objective (MO) problem have PLs and the LBI as the main parameters that need to be optimized. For that purpose, the MOGFPA is proposed in this paper. The MOGFPA consists of a golden search (GS) and tangent flight with Pareto distribution that only needs a few tuning parameters. Therefore, it is simple to alter these parameters to reach the best values compared to other existing methodologies. Its performance is predicted using different case studies on multiple test bus systems, namely the IEEE systems such as 33, 69, 119, and Indian 52 bus. Through simulation outcomes, the MOGFPA computes the optimum distribution of DG units and reconfigures the DNs with the aim of minimal PLs and LBI. Furthermore, another state-of-the-art technology and comparing convergence charts provide optimal outputs in less time, with minimum iterations.

**Keywords:** distribution network; load balance index; meta-heuristics; multi-objective; power loss minimization; sustainability

**MSC:** 90C27

## 1. Introduction

Due to the sustainable resources and government subsidies, there has been a flow of interest in different sources of renewable energy (RE), such as biomass, wind, and solar energy, throughout the world [1]. RE accounted for 16.7% of the worldwide energy consumption (EC) in 2010. The solar photovoltaic (PV) source of energy grew at the fastest rate of all RE sources, with an annual growth of 58% from late 2006 to 2011; in 2012, PV energy reached slightly more than 102 GW of worldwide installed capacity [2]. In 2017, this quantity was predictable to obtain more than 420 GW. Depending on the selected PV technology and location, a power system (PS) may withstand up to 50% of PV penetration [3]. The policies and market implementation on PV have been measured from 2019 to 2021 [4]. Multiple impact guides of a PV installation on climate change consequences are surveyed until 2023 [5]. On the other hand, time-dependent load models may have various implications on PV penetration predictions [6].

On account of planning for distributed generation (DG), multi-objective (MO) [7] optimization considers reactive and active power losses (PLs), as well as voltage variations. While such planning [8] studies are suitable for deploying dispatchable resources such as gas turbines, they have yet to address a real-world scenario that includes fluctuating demand and non-dispatchable RE. There have been a few recent research investigations on sustainable DG grid integration for PL reduction, all of which consider the dynamic load

variation for time. By adopting a genetic algorithm (GA) for loss reduction, the DG size was determined [9]. However, evolutionary-based algorithms are selected to investigate the optimal size of sources based on the manufacturing technologies [10]. The probabilistic analytics-based algorithms are proposed to allocate the optimal place for DG in DNs [11]. However, the impact of voltage and power profile was not measured.

Compared to constant load models, a few recent surveys have demonstrated that dependent models based on voltages substantially impact DG penetration planning. On the other hand, such research works predicted that DG units may be sent and distributed based on peak load demand. According to research work [12], it has been observed that dynamic load variation models affect the optimal allocation of DG (ADG). However, the authors did not discuss the effect of non-dispatchable sources on the ADG.

Nowadays, power usages are moving toward renewable energy resources (RERs) to fulfil the energy demand for different industrial sectors [13]. The change to more ecologically approachable and cost-effective mobility networks is being powered by growing global environmental tests, rising oil prices, and developing industrial criteria [14]. As a result, the most exciting means of mobility have emerged as electric vehicles (EVs), hybrid electric vehicles (HEVs), and plug-in electric vehicles (PEVs). It has been noted that, when compared to traditional home domestic loads, EV charging loads are rather large. For that purpose, an advanced grid structure comprising high-voltage direct current (HVDC) power lines, energy storage systems (ESSs), and flexible AC transmission (FACTS) systems are used.

In today's fast-growing world, the rapid integration of EV loads and their remote mobile connections to smart grids (SGs) has increased energy consumption to the highest level and needs more attention for reliable operation. Furthermore, if many EV fleet users returning home connect their EVs to the SG system and charge their cars during peak demand hours when the conventional load is also charging, a worst-case scenario may occur. Due to the increased number of EVs, PV systems, and non-uniform load profiles, network control and PS management are more complex.

Moreover, unregulated EV charging overloads the PS network and causes voltage violations, higher PL, and poor network management. As a result, developing a smart EV charging management system meets the consumers' charging demands while not jeopardizing SG objectives. Under a centralized control model, smart scheduling techniques empower the DG operator to make charging decisions that balance the grid and consumer interests. The evolution of intelligent and modern SG provides a solid foundation for using centralized scheduling systems (CSSs). Furthermore, smart load management for EVs leads to high-tech applications and economic flexibility, making consumer demand-side management (DSM) more relevant.

Several studies have zoomed in on the effects of EV charging and its scheduling tactics. One study [15] used the GA optimisation method to construct thoughtful EV planning based on the minimal load variation of sub-station transformers. Integrating EVs in valleys to provide smooth load demands has effectively decreased load stress on the system. Correspondingly, to handle the clustering of EVs in a power distribution system (PDS), researchers [16] conducted a planning-level study. The scheduling work is accepted using GA to lower system costs and emissions. In previous work [17,18], the researchers could charge EVs. They completed the challenging scheduling project using linear programming while optimizing the usage of RE sources. Another research [19] proposed control operations of EV load profile management to advance energy and costs by boosting power provided to the EV batteries; however, that paper failed to account for real domestic network restrictions. Furthermore, an earlier study [20] advised that EV charging be timed as efficiently as possible to reduce network PLs.

It has been observed from the literature that optimal DG placement with feeder routing with a volatile load bound to time is not being discussed. This research aims to find an optimal location and route in DNs while minimizing the load balance index (LBI) and PLs.

The LBI is an analytical equation with real and reactive power profile indices. It makes the scenario an MO problem.

In the early decades, different tests were conducted by adopting optimization techniques based on conventional methods such as the interior point method [21], the GS method [22], and meta-heuristic-based algorithms. The optimal solution is achieved without falling into local optimal points [23]. Metaheuristic algorithms (MAs) are classified based on the source of inspiration: evolutionary algorithms based on discretization concepts, physical models, swarm-dependent, and human inference-based algorithms. The first type, referred to as evolution algorithms, comprises specific steps such as reproduction, mutation, and offspring selection as some of the main stages. The most broadly used population evolutionary methods for wide-ranging optimization difficulties include genetic algorithms (GAs), evolution strategy (ES), genetic programming (GP), and other named techniques that are found in the literature.

The optimization algorithms that work on mimicking animal social behaviour include particle-swarm-optimization (PSO), cuckoo search optimization (CSO) [24], and cat swarm algorithm (CSA) [25]. Furthermore, different hybrid techniques such as chaotic PSO (CPSO) [26] and chaotic artificial bee colony (CABC) [27] are found in detail, and a comparative analysis has also been performed.

The modified flower pollination algorithm (MFPA) [28] addressed all of the issues mentioned above by developing a computational formula based on hybridizing many advanced methods to allow for adaption while searching for results to the optimization problem. This updated method surpasses the traditional one by incorporating all 23 well-known unimodal and multimodal test systems and 27 non-linear equation systems. Furthermore, compared to competing algorithms, it still faults its exploring operator, which could prevent it from attaining higher results for particular test functions. Henceforth, a hybridized version of flower pollination algorithm (FPA) is considered to solve the MO problem.

Suitable for many differential evolutions depending on past success, success history-based adaptive multi-objective differential evolution (SHAMODE) [29], an MO variation based on the use of an external Pareto archive, is proposed and compared to numerous MO evolutionary algorithms. An external library is formed to gather the updated non-dominated solutions from the initial replication in an iterative manner. The algorithm has been improved to make it appropriate for MO optimization situations. This algorithm is reserved for comparative analysis.

Among all, the FPA is taken in this article. It is advanced to solve global optimization issues by simulating the pollination process of flowers and has proven to be effective in a wide range of optimization problems. However, it still suffers from local optimal stagnation because, during the optimization process, it was not able to elaborate on different regions within the search area, as well as due to its slow convergence rate, which forces the classical FPA to go through multiple iterations to find better alternatives in the areas that are unpromising in nature. In general, a mathematical test of about 10 for a population size of 25 and iterations of 10,840 was used to evaluate the FPA; this is assumed to be the reasonable consumption rate for reaching the required findings. In addition, to solve global test functions of about 23, the authors used a conventional FPA working with a clonal selection algorithm. In the case of MO problems, the Pareto optimal set is attained by a non-dominated ranking procedure after sorting the solutions.

For that purpose, the multi-objective golden flower pollination algorithm (MOGFPA) is proposed to solve such a problem while keeping the network parameter constraints under limits. The proposed method uses tangent flight and golden section search to address the issue of DG allocation in PDS. The GS algorithm can better predict local optimal spots on the dynamic nature of systems, whereas the tangent flight method is more capable of discovering the overall solution by probing the surroundings. In addition, the proposed method is used if it offers a solution based on the dynamism of the network and it modifies its configuration in response to tie switching. This algorithm will also determine the best path while keeping an eye on the PLs and LBI. The algorithm was tested under different

load-varying conditions in the residential, commercial, and industrial sectors. It was compared with cutting-edge methods such as multi-objective flower pollination algorithm (MOFPA) [30], SHAMODE, and a hybrid version of SHAMODE with whale optimization (SHAMODE–WO) [31] to further demonstrate its novelty. The comparison is based on convergence charts showing that our proposed algorithm converges faster than other algorithms, providing an optimal solution in less time. From our perspective, it is the first study in FPA that combines GS and tangent flight to solve a complex MO problem. Testing the proposed MOGFPA method on IEEE systems such as 33, 69, 119, and Indian 52 bus test systems validated its efficiency.

*The primary purpose of this work is:*

- To reduce PLs and the LBI, the study utilizes a MOGFPA to determine the ideal reconfiguration after locating a place for ADG;
- MOGFPA is designed with fewer parameters, which minimizes system complexity and enhances system dependability;
- The research focuses on optimally determining the solution for complex MO functions in minimum time and fewer iterations.

The remaining paper is sequenced as the proposed research work, which comprises problem formulation in Section 2 and MOGFPA optimization, which is described in Section 3. The simulation findings and observations based on standard test bus setups are the main topics of Section 4. In the end, Section 5 is used to conclude the paper.

## 2. Problem Formulation

The proposed study's initial objective is to create an effective optimization method for choosing a suitable feeder location for PDS with optimal ADG. Using the MOGFPA, this research offers a novel system for determining the best DG position and routing. The objective is to reduce the LBI and PLs. Using test systems for the IEEE systems such as 33, 69, 119-bus, and practical Indian 52 bus, the proposed algorithm is proven.

### 2.1. Modelling the Electrical Load

The power demand by the end users typically follows the load curve. The load of factor (LoF) for a PS under different load conditions is expressed in Equation (1) below.

$$LoF = \sum_{T=1}^{24} \frac{pu\ demand}{24} \qquad (1)$$

where *pu* demand is per unit demand, and due to dynamic load variation in voltage profile, the power profile equation at period (t) may be represented as shown in Equation (2).

$$Powk(t) = Pok(t)xVk(t), \quad Qowk(t) = Qok(t)xVk(t) \qquad (2)$$

where:

$Pok(t)$: active power that is generated at bus '$k$';
$Qok(t)$: reactive power that is generated at bus '$k$';
$Powk(t)$ and $Qowk(t)$: at nominal voltage, bus '$k$' has a dynamic and reactive load;
$Vk(t)$: voltage at bus $k$.

### 2.2. Solar Modelling

Based on three years of historical data, the probability density beta function (PDeF) is used to replicate solar irradiance for each hour of the day. For this PDeF, one day is classified into 24 h intervals, each of which has its distinct solar irradiance PDeF and lasts for one hour. The average and variance of the day's hourly solar irradiance are determined from historical data. A 0.05 kW/m$^2$ step separates each of the 20 sun irradiance levels that make up each hour. The mean and deviation are used to construct a PDeF with 20 different solar irradiance states for a whole day, and the likelihood has been calculated. As a result,

the hour's PV power output is computed. The model's description is found below. The PDeF reported [32] shows the probabilistic solar irradiation, as in Equation (3).

$$F(s) = \{ \frac{\tau(\alpha + \beta)}{\tau(\alpha)\tau(\beta)} So^{(\alpha-1)}(1 - So)^{(\beta-1)}, \ 0 < So < 1, \ \alpha, \beta > 0 \tag{3}$$

where $F(s)$ denotes the beta probability distribution for PV radiance, and $\tau$ indicates the probability density function. The terms '$\alpha$' and '$\beta$' are the parameters of $F(s)$. They were derived from its mean and standard deviation. The term '$So$' indicates the random variable.

According to the current IEEE 1547 standard [33], for the grid, this work requires reactive power. But PV inverters are not allowed for this, which causes a lack of reactive power reward for voltage control because the PDS is modified with significant PV penetration and only active power boosters. Voltage regulation is provided by some slow devices such as voltage regulators, capacitors, and tap changers, but these devices are slow and cannot compensate for the temporary measures made by PV intermittency. Undoubtedly, the absence of reactive power support will become a key challenge at the PDS. As reversing the flow of power tends to fluctuate with time and climate variations, the existence of surplus PV power and low demand at the same time may result in voltage regulation ill effects and unanticipated voltage spikes on feeders [34]. According to the new German SG [15], inverter-equipped PV units absorb energy to stabilize load voltages. It is considered a rapid response energy device while also producing energy as a primary goal where the PV unit generating reactive power is positive and the PV unit consuming reactive power is opposing.

### 2.3. Load Flow Analysis

Power quality is determined mainly by how well PDSs reduce active PLs. The standard PDS will install DG units to lower active PLs. The power quality, adaptability, reliability, and efficiency of the PDS are significantly impacted by the load flow technique used to address the issue of ADG. To expand these networks' performance, the correct optimization technique is applied. Load flow analysis, a crucial step in the optimization process, determines the optimisation algorithms' potential and accuracy. This study examines the distribution network's power flow using a forward–backward iterative method, and the best ADG units are measured from the resulting solution.

### 2.4. Objective Functions

The goal functionality is proposed to decrease PLs while simultaneously decreasing the distribution system's LBI. The goal function identified in this study is as follows in Equation (4).

$$O.F1 = minimize \sum_{i=1}^{n} PL \tag{4}$$

where 'n' is the total number of nodes, PL denotes the active power loss in each node, and O.F denotes the objective function. Active PL is measured using the formula below, as shown in Equation (5):

$$PL = \sum_{j=1}^{n} I_j^2 R_j \tag{5}$$

where $I_j$ denotes the amperage of the current and $R_j$ denotes the node-specific resistance in ohms.

This paper uses an optimization algorithm that would keep the LBI in check to keep the PDS stable by finding optimal feeder reconfiguration. The system is reconfigured, so

all lines carry an optimum current value. The LBI can express the second objective of this optimization problem in Equation (6).

$$O.F2 = minimize \sum_{j=1}^{n} LBI = \sum_{j=1}^{n} Lj \frac{Ij^2}{I^2} \tag{6}$$

where $Lj$ is the length of the individual branch, $Ij$ is the complex current flow in each line, and $I$ is the rated current of buses.

### 2.5. Constraints

The operational constraints for assessing the PLs and LBI in the PDS are labelled with Equations (7)–(9).

- Voltage limit constraints

$$V_{min} \leq V_i \leq V_{max} \tag{7}$$

where $V_i$ is the bus feeder's input voltage and $V_{min}$ and $V_{max}$ are the minimum and maximum voltage values permitted (10%), respectively. Voltage thresholds are noted and verified when the ADG and network shifting operations are in progress.

- Current limit constraints

In the feeder, the current flow should fall within the maximum limit.

$$0 \leq I_j \leq I_{max,j}; \ j = 1, \ldots, N^{BR} \tag{8}$$

where $I_{max,j}$ is the maximum flow of current in PDS, along with the number of branches denoted by $N^{BR}$, and $I_j$ is current passing in the branch '$j$'.

- Power balance limits

$$P_s + \sum_{k=1}^{N_G} P_G = P_d + PL \tag{9}$$

where $N_G$ stands for the total number of generator buses, $P_s$ for the slack bus power, $P_G$ denotes the power at generator buses, $P_d$ for the load power, and $PL$ is active power loss. When rerouting the flow via switch operation, the primary consideration is the overall $PL$, and the path with fewer losses is preferred.

## 3. Proposed Methodology

Significant progress has been made in developing several MO methods in recent decades. These MAs blend swarm intelligence and computational intelligence and are motivated by the biological behaviour of many species. The MAs are efficient, flexible, and simple to implement. The proposed study uses the MOGFPA to determine the ideal feeder connections position to decrease a PDS's PLs.

### 3.1. Multi-Objective Golden Flower Pollination Algorithm (MOGFPA)

A swarm-based optimization method inspired by flower pollination behaviour is the FPA method. Flowers replicate through a pollination process, in which pollinators spread one flower's pollen to another. Generally, the pollination process is classified into two types: *biotic and abiotic*. In biotic pollination, live organisms such as insects, birds, and other animals function as pollinators, whereas in abiotic pollination, pollen is transmitted by wind or diffusion, with no pollinator required.

Self-pollination and cross-pollination are the two types of pollination in flowers. When a flower replicates by using its pollen or the pollen of other flowers on the same plant, it is said to be self-pollinating. When pollen is transported across long distances by pollinators, cross-pollination takes place. Self-pollination is also referred to as local pollination, while

global pollination refers to pollination that occurs over a wide area. As a result of the pollen's higher movements and Levy flying behaviour during global pollination, this method is used to accomplish global pollination.

The implementation of an FPA for optimizing the process of ADG follows four rules. When the FPA creates a new population, the parameter p controls whether the population is made through cross-pollination or self-pollination [35].

This is performed by producing a random variable within a range of 0–1, comparing it to p, and assuming that global pollination will occur if the random number is less than p (local pollination occurs if the random variable is more significant than p).

The conventional FPA method employs Lévy flights in a global random walk to explore the search space. Lévy distribution, an extended tail probability distribution, is used to derive the Lévy step. In this situation, a percentage of giant steps is generated, which helps global search more effectively. Many migratory and foraging animals follow the Lévy distribution, but there has not been enough research on how other heavy tail probability distributions affect FPA. This motivates us to try to adapt the famous Mittag–Leffler, Pareto, Cauchy, and Weibull distribution to the basic algorithm to conduct more efficient searches. Pareto distribution is considered as follows.

If the cumulative distribution function of a random variable contains the following equation, it is said to be susceptible to Pareto distribution, as shown in Equation (10).

$$P(x) = \begin{cases} 1 - \left(\frac{b}{x}\right)^a, & x > b \\ 0, & x < b \end{cases} \tag{10}$$

where $b$ represents the scale and $a$ represents the shape parameter. $x$ indicates the fitness parameter.

We have used the alternative heavy-tailed probability distribution function in a modified algorithm in place of Lévy flights as practised in the global explorative process in the MOGFPA method [36].

In Equation (11), a random selection from the Pareto distribution is denoted by $P(x)$;

$$X_{t+1} = X_t + stp * P(x) * (X_t - g*) \tag{11}$$

where $X_{t+1}$ denotes solution vector at the $t+1$ iteration. $X_t$ denotes solution vector at the $t$ iteration. $P(x)$ is used to represent Pareto distribution. The $g*$ is the best global solution. Additionally, the $stp$ is the step size. The switching probability controls the local and global search processes in the FPA. The switching probability allows pollens to investigate and be used during global and local pollination operations. Pollens are permitted to explore the key space to identify the best method for global pollination while maintaining the diversity of the approaches. The exploration and operation phase of the FPA allows it to perform better than other algorithms and handle complex situations.

On the other hand, the FPA deviates from the optimal solution due to the random nature of these two phases, dependent on the switching probability. In order to attain the best outcome in the least amount of time for the best placement of DG and reconfiguration networks, this paper employs a hybrid optimization technique. The FPA tangent flight method and the golden-section search (GS) algorithm make up the advanced hybrid algorithm. Through GS, the best local solution for local pollination is accomplished, while tangent flight provides the best global solution. This way, the problem is solved whether it has a local optimal or global solution, as shown in Figure 1.

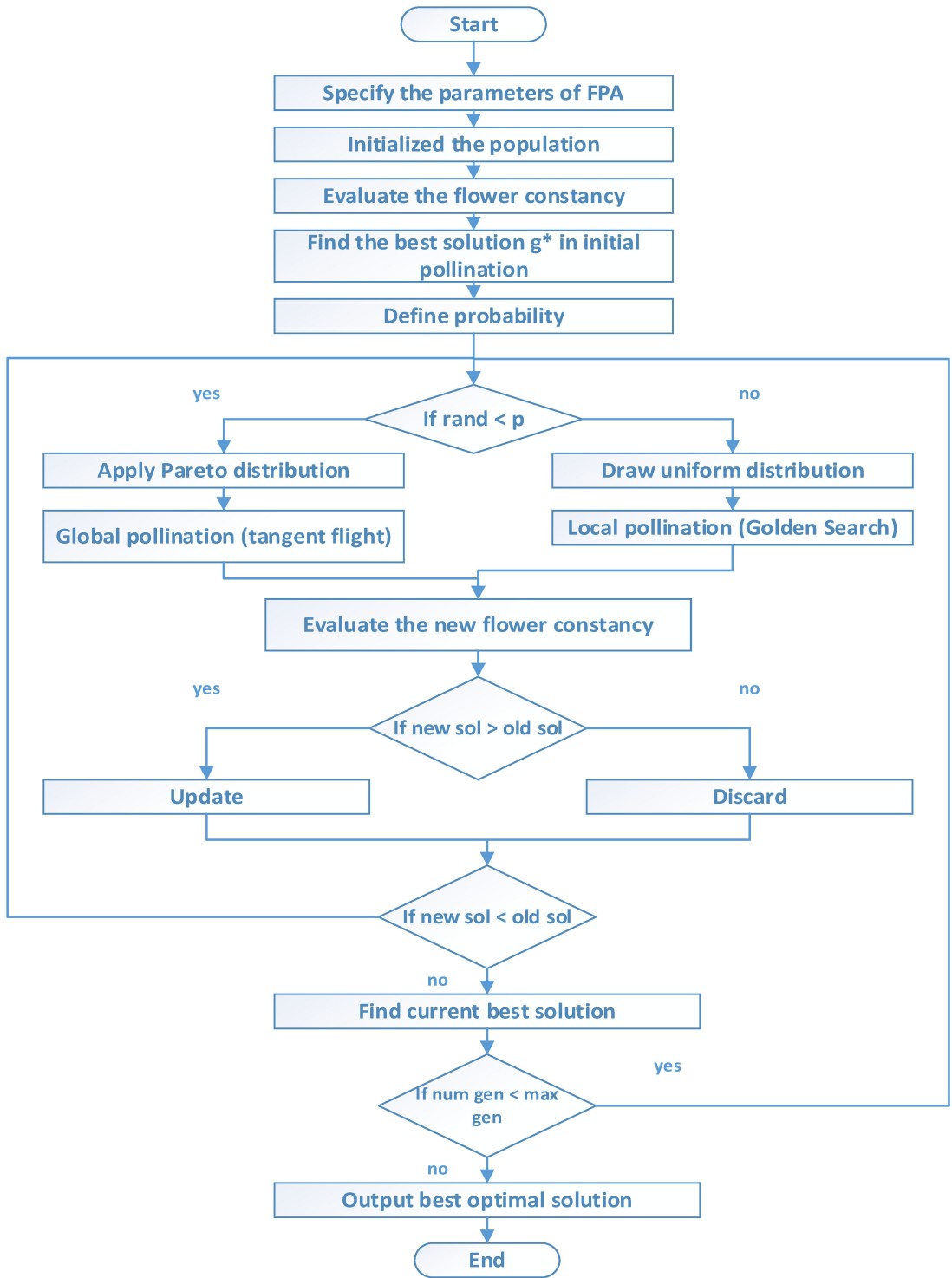

**Figure 1.** Flow chart of the proposed algorithm.

### 3.2. GS Algorithm

The phrase "Golden Section" refers to the renowned problem of properly dividing line segments. The line segment is divided into two segments, L1 and L2, which equal significant and minor line lengths of L search space, respectively [22].

Φ is the golden ratio in Equations (12) and (13):

$$\Phi = \frac{1+\sqrt{5}}{2} = 1.618 \tag{12}$$

$$\alpha = \frac{1}{\Phi} = 0.618 \tag{13}$$

The inverse of the golden ratio ($\alpha$) is the golden section, commonly referred to as the ratio of minor-to-significant subsegments.

Equations (14) and (15) are the two separate equations from the search space:

$$X_1 = a + 0.618(b - a) \tag{14}$$

$$X_2 = b - 0.618(b - a) \tag{15}$$

where $X_1$ and $X_2$ are two solution vectors. $a$ and $b$ are minor and major segments, respectively. The most significant point's abscissa cannot be smaller than $X_1$. As a result, the highest value will lie within the $[X_1, b]$ range, and vice versa. This new interval is applied for the subsequent iteration.

### 3.3. Tangent Flight Algorithm

This study's main objective is to provide a direct optimization method. The tangent function, a fundamental mathematical operation, is the foundation for the tangent flight algorithm (TFA) [37]. The frequency and oscillation between -infinity and +infinity provide an excellent balance of exploration and intensification, and this function has a significant amount of exploration potential. The equation of motion in the TFA technique is driven by a global step of the form "stp*tan()," where the tangent function behaves similarly to the Levy flight function. For convenience, this work calls it a tangent flight. Most optimization techniques, including the following, are based on a descent Equation (16).

$$X_{t+1} = X_t + stp(d) \tag{16}$$

where $X_{t+1}$ and $X_t$ are solution vectors at $t + 1$ and $t$ iteration. $stp$ is the size that needs to move, and $d$ is the direction. While derivative-based approaches calculate this step using gradient or Hessian information, free derivative methods—such as metaheuristics—use stochastics to converge to global optima. For example, the step size is measured by Gaussian mutation in a GA. The step size in differential evolution is estimated by the difference between individuals in the present population, whereas the step size is approximated using a Levy flight function in FPA, and so on.

$$X_0 = lwb + (upb - lwb). * rand(D) \tag{17}$$

where $X_0$ is the solution vector obtained through the intensification process. The lower and upper limits are $lwb$ and $upb$, respectively, while the size of the optimization problem is D. Too much intensification in Equation (17) causes the program to lag and occasionally depart from a local minimum, whereas too much study causes the program to lag and sometimes diverge. Tangent flight accomplishes this goal in three ways: intensification, exploration, and escaping local minima. The search variables are explored to identify the promising options in total space. On the other side, the intensification phase is used to guide for the best alternative among the population. Finally, the escape local minima method is applied to a random search agent (solution) at each loop to avoid becoming stuck in a local minimum. Similar to some other population-based optimization techniques, the TFA starts by producing a random starting population within the solution space. The initial solution is distributed evenly over the search space using Equation (18).

$$X_{t+1} = X_t + stp * \tan\theta * (X_t - g*) \tag{18}$$

where solution vectors at $t + 1$ and the $t$ iteration are represented by $X_{t+1}$ and $X_t$, respectively. $stp$ is the size that needs to move. $g*$ represents the best global solution. A random local walk uses the readings of the subsequent variable in the optimal solution to replace certain variables in the acquired solution. In optimization algorithms with dimensions

greater than 4, the percentage of replacement variables is equal to 20%, while in problems with dimensions lesser than 4, it is equal to 50%. The X value is reordered if its value exceeds the boundary condition.

In contrast to local search methods, the global random walk gives the metaheuristic-based population much room for exploration. This strategy creates a globally random walk due to the combination of variable step size and tangent flight. The tangent function aids in the practical exploration of the search space. Indeed, tangent values around pi/2 will be more prominent, and the produced solution will be far from the proposed solutions, while tangent values near 0 will be small, and the obtained solution will be close to the real system. As a result, the exploratory search equation combined the global and local random walks.

The TFA has a method to address this issue by following a precise procedure. The operation is divided into two parts, each of which is accepted with a certain degree of probability. One agent search is selected randomly in each iteration, and one of the above equations is shown. Furthermore, a new, unexpected solution with a chance of 0.01 can replace the worst response to solve the dilemma of local minimum stagnation.

## 4. Results and Discussions

The designed technique is validated and justified in this section. The intended method is approved in MATLAB 2020, and the results are measured. Concerning IEEE systems such as 33, 69, 119, and the Indian 52 bus system, the proposed method is tested. A 4GHz Intel Core i7 machine and 32 GB of RAM are used to complete the due process. The proposed approach aims to minimize the PLs of the RDS system by improving its feeder connections following the placement of the DG. This ideal network reconfiguration is calculated with a novel idea developed by the MOGFPA, RDS, and it is used to determine the best tie switch connections, which reduces the PLs and evaluates the LBI considering MO. The whole problem is viewed as an MO problem, having PLs and the LBI as primary parameters. The planned technique is compared with current approaches such as SHAMODE [29], MOFPA [30], and SHAMODE–WO [31]. The main objective is to configure the best location and power routing part for DG while keeping the constraint imposed in check.

Moreover, the optimal routing configuration is completed by monitoring PLs and the LBI in review. Different load types are computed concerning energy exponent values [12], as depicted in Figure 2. These exponent values are incorporated along with specified loads to obtain a dynamic load profile, as shown in Figure 3. The configuration for the test system is summarized in Table 1.

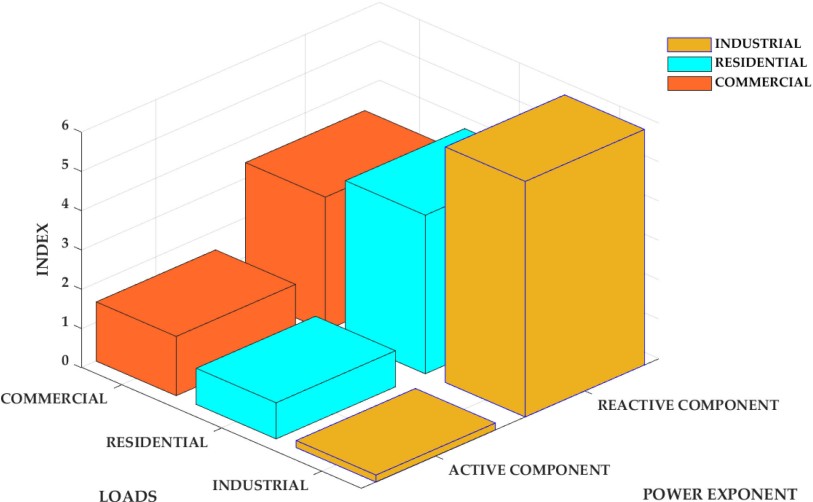

**Figure 2.** Load types with power exponents.

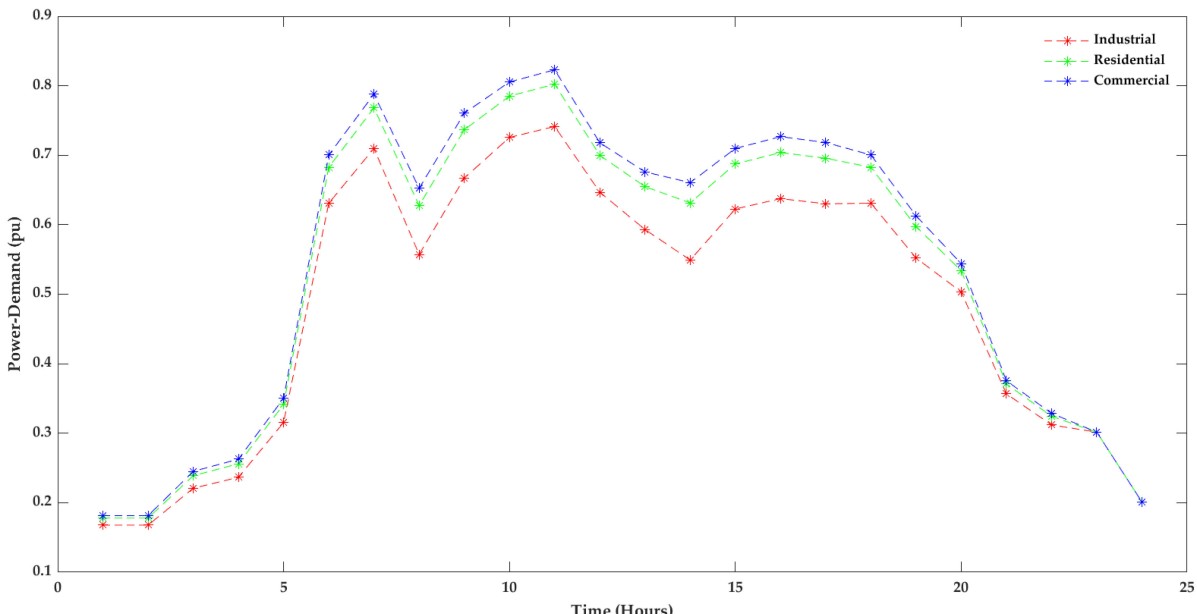

**Figure 3.** Different load profiles for systems are taken into consideration.

**Table 1.** Specifications of the test systems.

| | Base kV | Base MVA | Tie Switches | DG Location | DG Power (kW) | Bus Location for ESS | ESS Power (kW) |
|---|---|---|---|---|---|---|---|
| IEEE 33 bus | 12.66 | 100 | 5 | 9, 13, 29 | 396, 555, 567 | 18 | 30 |
| IEEE 69 bus | 12.66 | 100 | 5 | 1, 12, 61 | 918, 266, 206 | 27 | 30 |
| IEEE 119 bus | 11 | 100 | 15 | 34, 71, 111 | 1626, 1858, 1226 | 55 | 30 |
| Indian 52 bus | 11 | 1 | 6 | 11, 35, 45 | 476, 272, 472 | 26 | 30 |

*4.1. Evaluation of IEEE Systems*

The IEEE 33 bus system has been considered for visualizing the performance of the proposed MOGFPA. Through the MOGFPA, the average PLs and LBI are determined to be 282.86 kW, 0.0015, 281.76 kW, 0.0014, 284.24 kW, and $1.49 \times 10^{-3}$ for industrial, residential, and commercial loads, respectively. That is much lower than other algorithms such as SHAMODE [29], MOFPA [30], and SHAMODE–WO [31]. The bus system PLs are reduced with the help of the above methodology. The proposed algorithm is used to optimally select the optimal location for feeder connections and DG while considering the problem of MO having PLs and the LBI as crucial parameters. The optimal switching connections while implementing the algorithms are summarized in Table 2. This means that by determining the best point for DG placement by minimal PLs and determining the optimal configuration for switches, the parameter LBI also needs to be addressed carefully. Moreover, the minimum and maximum limits are also compared, as shown in Table 3, to show the robustness of the proposed algorithm.

**Table 2.** Tie switches configuration for IEEE 33 bus system.

| | Industrial | | | Residential | | | Commercial | | |
|---|---|---|---|---|---|---|---|---|---|
| | Average (Aver) | Min | Max | Aver | Min | Max | Aver | Min | Max |
| SHAMODE–WO | 33–9 | 16–12 | 11–26 | 16–11 | 10–6 | 15–29 | 10–6 | 26–33 | 6–18 |
| | 15–20 | 3–33 | 17–12 | 33–15 | 26–13 | 10–33 | 13–16 | 13–5 | 27–17 |
| | 17–11 | 27–26 | 33–15 | 17–27 | 16–17 | 27–26 | 27–11 | 17–16 | 26–11 |
| | 3–29 | 5–15 | 10–3 | 22–3 | 23–14 | 13–8 | 5–26 | 12–21 | 8–22 |
| | 13–12 | 13–11 | 31–6 | 9–29 | 27–20 | 17–23 | 15–18 | 9–20 | 7–9 |
| SHAMODE | 17–5 | 11–27 | 10–16 | 17–15 | 10–27 | 33–11 | 11–27 | 20–6 | 15–27 |
| | 10–33 | 9–19 | 12–33 | 27–26 | 12–16 | 10–16 | 17–33 | 33–17 | 26–17 |
| | 26–9 | 6–12 | 5–27 | 9–2 | 11–5 | 15–6 | 26–15 | 10–13 | 6–13 |
| | 16–15 | 23–33 | 13–2 | 29–5 | 33–6 | 31–2 | 12–10 | 26–9 | 3–11 |
| | 21–6 | 16–13 | 6–15 | 33–12 | 20–19 | 17–5 | 20–22 | 18–5 | 16–29 |
| MOFPA | 4–31 | 10–21 | 9–25 | 16–33 | 12–23 | 15–19 | 8–12 | 8–26 | 5–19 |
| | 27–17 | 29–12 | 33–16 | 32–19 | 17–15 | 4–29 | 14–16 | 21–23 | 25–21 |
| | 24–13 | 26–6 | 32–31 | 28–14 | 26–6 | 8–10 | 6–10 | 15–31 | 7–4 |
| | 18–26 | 17–16 | 18–27 | 17–25 | 8–14 | 28–7 | 33–9 | 30–12 | 30–20 |
| | 25–29 | 25–8 | 2–15 | 27–3 | 4–32 | 20–18 | 2–30 | 6–3 | 16–29 |
| MOGFPA | 13–2 | 4–21 | 5–31 | 20–4 | 31–3 | 14–13 | 8–31 | 12–6 | 24–5 |
| | 8–15 | 2–9 | 32–19 | 27–25 | 18–28 | 16–22 | 30–15 | 4–7 | 15–16 |
| | 30–19 | 3–15 | 14–20 | 18–7 | 8–7 | 4–8 | 19–22 | 11–23 | 27–23 |
| | 24–28 | 17–13 | 12–23 | 14–22 | 29–32 | 10–32 | 26–33 | 18–19 | 2–12 |
| | 27–33 | 14–7 | 18–29 | 21–33 | 2–4 | 15–9 | 3–21 | 32–3 | 17–13 |

**Table 3.** PL and LBI comparison under different load conditions for the IEEE 33 bus system.

| LOADS | | | SHAMODE–WO | SHAMODE | MOFPA | MOGFPA |
|---|---|---|---|---|---|---|
| Industrial | Aver | PL (kW) | 294.61 | 294.61 | 283.02 | 282.86 |
| | | LBI | 0.044 | 0.04 | 0.0016 | 0.0015 |
| | Min | PL | 293.32 | 293.32 | 284.44 | 282.9 |
| | | LBI | $1.68 \times 10^{-3}$ | $1.68 \times 10^{-3}$ | $1.59 \times 10^{-3}$ | $1.51 \times 10^{-3}$ |
| | Max | PL | 296.13 | 296.13 | 286.67 | 284.24 |
| | | LBI | 0.016 | 0.013 | 0.0014 | 0.0014 |
| Residential | Aver | PL | 286.54 | 286.54 | 283.19 | 281.76 |
| | | LBI | 0.037 | 0.04 | 0.0015 | 0.0014 |
| | Min | PL | 286.03 | 286.03 | 283.57 | 280.66 |
| | | LBI | $1.69 \times 10^{-3}$ | $1.82 \times 10^{-3}$ | $1.53 \times 10^{-3}$ | $1.47 \times 10^{-3}$ |
| | Max | PL | 283.42 | 283.42 | 286.67 | 282.9 |
| | | LBI | $1.29 \times 10^{-2}$ | $1.67 \times 10^{-2}$ | $1.59 \times 10^{-3}$ | $1.51 \times 10^{-3}$ |
| Commercial | Aver | PL | 300.83 | 300.83 | 286.67 | 284.24 |
| | | LBI | $3.16 \times 10^{-2}$ | $3.20 \times 10^{-2}$ | $1.49 \times 10^{-3}$ | $1.49 \times 10^{-3}$ |
| | Min | PL | 289.04 | 289.05 | 283.09 | 282.56 |
| | | LBI | $1.76 \times 10^{-3}$ | $1.83 \times 10^{-3}$ | $1.63 \times 10^{-3}$ | $1.48 \times 10^{-3}$ |
| | Max | PL | 310.74 | 310.74 | 284.44 | 289.36 |
| | | LBI | $1.36 \times 10^{-2}$ | $1.39 \times 10^{-2}$ | $1.69 \times 10^{-3}$ | $1.55 \times 10^{-3}$ |

The proposed MOGFPA is assessed on the IEEE 69 bus system to demonstrate the generalizability of the algorithm. The average PLs and LBI are determined to be 31,281.31 W, $1.28 \times 10^{-2}$, 28,170.69 W, $6.22 \times 10^{-4}$, 28,275.68 W, and $8.20 \times 10^{-4}$ for industrial, residential, and commercial loads, respectively. That is much lower than other algorithms such as SHAMODE [29], MOFPA [30], and SHAMODE–WO [31]. The bus system PLs are reduced with the help of the planned methodology. The proposed algorithm is used to optimally select the optimal location for feeder connections and DG while considering the problem as a MO having PLs and the LBI as crucial parameters. The optimal switching connections while implementing the algorithms are summarized in Table 4. This means that by only determining the best point for DG placement, the PLs cannot be handled, and choosing the optimal configuration for switches is the crucial parameter that needs to be addressed carefully. Moreover, the minimum and maximum limits are also compared, as shown in Table 5, to show the robustness of the proposed algorithm.

**Table 4.** Tie switches configuration (IEEE 69).

|  | Industrial | | | Residential | | | Commercial | | |
|---|---|---|---|---|---|---|---|---|---|
|  | Aver | Min | Max | Aver | Min | Max | Aver | Min | Max |
| SHAMODE–WO | 63–53 | 32–15 | 41–35 | 42–41 | 38–5 | 20–19 | 35–60 | 67–58 | 41–67 |
|  | 14–32 | 26–5 | 5–57 | 31–5 | 14–3 | 63–25 | 38–15 | 15–30 | 56–19 |
|  | 56–43 | 43–63 | 53–25 | 53–32 | 25–41 | 39–41 | 52–58 | 32–31 | 23–43 |
|  | 47–42 | 35–31 | 4–66 | 63–30 | 2–23 | 47–22 | 56–42 | 47–4 | 32–63 |
|  | 2–35 | 23–2 | 27–3 | 60–38 | 30–33 | 57–43 | 43–32 | 56–3 | 15–6 |
| SHAMODE | 38–35 | 39–33 | 66–56 | 20–33 | 22–63 | 57–53 | 58–26 | 22–6 | 34–2 |
|  | 3–33 | 23–43 | 22–19 | 34–31 | 56–28 | 3–6 | 14–52 | 14–20 | 20–57 |
|  | 22–58 | 60–20 | 26–60 | 63–15 | 41–42 | 30–33 | 35–41 | 19–43 | 63–55 |
|  | 15–13 | 42–6 | 13–57 | 13–23 | 58–34 | 4–67 | 63–27 | 40–2 | 19–33 |
|  | 32–34 | 31–5 | 5–63 | 57–14 | 15–52 | 13–15 | 43–5 | 63–38 | 4–27 |
| MOFPA | 32–40 | 52–10 | 62–15 | 50–57 | 45–30 | 56–23 | 20–48 | 37–56 | 65–26 |
|  | 22–34 | 9–33 | 65–45 | 22–54 | 41–34 | 26–66 | 10–6 | 45–49 | 30–14 |
|  | 35–6 | 41–38 | 39–69 | 43–17 | 21–57 | 10–19 | 23–54 | 9–33 | 46–21 |
|  | 55–3 | 69–2 | 6–61 | 14–51 | 66–46 | 32–46 | 49–38 | 28–67 | 12–61 |
|  | 68–16 | 17–59 | 33–24 | 61–20 | 44–42 | 31–18 | 2–4 | 29–35 | 68–52 |
| MOGFPA | 8–32 | 18–4 | 15–61 | 35–68 | 37–57 | 4–15 | 39–31 | 37–57 | 65–37 |
|  | 61–60 | 35–33 | 34–39 | 40–11 | 20–14 | 51–24 | 32–12 | 20–14 | 63–47 |
|  | 42–45 | 20–9 | 30–5 | 10–9 | 64–46 | 30–9 | 4–68 | 64–46 | 9–61 |
|  | 47–33 | 27–3 | 17–22 | 18–44 | 55–43 | 55–57 | 3–60 | 55–43 | 7–38 |
|  | 64–29 | 6–67 | 59–40 | 26–12 | 66–42 | 16–23 | 52–66 | 66–42 | 49–30 |

**Table 5.** PL and LBI comparison under different load conditions (IEEE 69).

|  |  |  | SHAMODE–WO | SHAMODE | MOFPA | MOGFPA |
|---|---|---|---|---|---|---|
| Industrial | Aver | PL(W) | 31,282.31 | 31,282.31 | 31,282.31 | 31,281.31 |
|  |  | LBI | $1.35 \times 10^{-2}$ | $1.30 \times 10^{-2}$ | $1.35 \times 10^{-2}$ | $1.28 \times 10^{-2}$ |
|  | Min | PL | 7791.94 | 7792 | 7791.9 | 7790.8 |
|  |  | LBI | $4.84 \times 10^{-3}$ | $4.84 \times 10^{-3}$ | $2.93 \times 10^{-3}$ | $1.07 \times 10^{-3}$ |
|  | Max | PL | 65,062.67 | 65,062.67 | 65,061.88 | 65,060.67 |
|  |  | LBI | $1.65 \times 10^{-2}$ | $1.65 \times 10^{-2}$ | $1.64 \times 10^{-2}$ | $1.63 \times 10^{-2}$ |

**Table 5.** *Cont.*

| | | | SHAMODE–WO | SHAMODE | MOFPA | MOGFPA |
|---|---|---|---|---|---|---|
| Residential | Aver | PL | 28,172.68 | 28,172.68 | 28,172.93 | 28,170.69 |
| | | LBI | $3.87 \times 10^{-2}$ | $1.37 \times 10^{-2}$ | $7.22 \times 10^{-4}$ | $6.22 \times 10^{-4}$ |
| | Min | PL | 8851.93 | 8852.94 | 8851.13 | 8850.93 |
| | | LBI | $2.95 \times 10^{-2}$ | $5.51 \times 10^{-3}$ | $5.21 \times 10^{-3}$ | $5.13 \times 10^{-3}$ |
| | Max | PL | 90,213.13 | 90,213.13 | 90,213.13 | 90,203.14 |
| | | LBI | $1.51 \times 10^{-2}$ | $2.97 \times 10^{-3}$ | $2.68 \times 10^{-2}$ | $2.79 \times 10^{-3}$ |
| Commercial | Aver | PL | 28,277.51 | 28,277.52 | 28,277.52 | 28,275.68 |
| | | LBI | $2.64 \times 10^{-2}$ | $9.64 \times 10^{-4}$ | $8.67 \times 10^{-4}$ | $8.20 \times 10^{-4}$ |
| | Min | PL | 9185.89 | 91,85.89 | 9185.9 | 9180.89 |
| | | LBI | $2.66 \times 10^{-2}$ | $2.18 \times 10^{-3}$ | $2.13 \times 10^{-3}$ | $2.12 \times 10^{-3}$ |
| | Max | PL | 99,331.01 | 99,331.02 | 99,331.24 | 99,301.03 |
| | | LBI | $5.23 \times 10^{-2}$ | $1.75 \times 10^{-2}$ | $5.54 \times 10^{-3}$ | $5.25 \times 10^{-3}$ |

To further show the robustness of the proposed algorithm, it was tested under a more extensive network IEEE 119 bus system, and the comparative parameters are analysed and summarized in Table 6. Moreover, the comparative analysis is visualized in Figures 4–6. This shows that the proposed algorithm outclassed the other algorithms by a significant margin in terms of average LBI, such as $1.09 \times 10^{-6}$, $1.08 \times 10^{-6}$, and $1.16 \times 10^{-6}$ for industrial, residential and commercial loads respectively.

**Table 6.** Tie switches configuration (IEEE 119).

| | Industrial | | | Residential | | | Commercial | | |
|---|---|---|---|---|---|---|---|---|---|
| | **Aver** | **Min** | **Max** | **Aver** | **Min** | **Max** | **Aver** | **Min** | **Max** |
| SHAMODE–WO | 30–55 | 47–83 | 51–62 | 56–64 | 17–15 | 39–40 | 99–38 | 89–75 | 41–18 |
| | 106–19 | 2–16 | 88–10 | 24–62 | 51–101 | 21–6 | 57–116 | 12–26 | 87–27 |
| | 95–104 | 81–44 | 78–49 | 54–100 | 5–72 | 4–26 | 5–67 | 95–2 | 35–69 |
| | 84–72 | 62–8 | 94–9 | 98–9 | 91–114 | 17–85 | 11–118 | 18–27 | 47–33 |
| | 92–26 | 82–105 | 64–48 | 75–104 | 44–23 | 104–83 | 69–40 | 67–17 | 90–62 |
| | 18–93 | 29–11 | 117–85 | 94–15 | 99–19 | 106–72 | 112–119 | 21–103 | 25–54 |
| | 17–52 | 99–88 | 15–67 | 105–69 | 58–104 | 18–56 | 51–66 | 11–59 | 7–3 |
| | 112–78 | 114–4 | 34–79 | 25–76 | 21–25 | 54–99 | 16–46 | 106–110 | 105–12 |
| | 76–100 | 68–67 | 53–100 | 6–114 | 47–4 | 49–108 | 15–59 | 14–87 | 19–6 |
| | 68–116 | 9–96 | 69–33 | 42–55 | 87–14 | 69–89 | 10–30 | 73–29 | 67–114 |
| | 9–4 | 92–57 | 61–81 | 48–72 | 79–22 | 94–15 | 60–3 | 58–39 | 43–13 |
| | 87–8 | 26–35 | 105–29 | 14–73 | 57–110 | 98–77 | 2–88 | 53–7 | 60–48 |
| | 110–27 | 118–24 | 46–95 | 19–60 | 92–18 | 46–116 | 81–85 | 100–46 | 22–83 |
| | 73–48 | 31–15 | 7–91 | 81–12 | 62–106 | 51–43 | 92–105 | 72–20 | 53–88 |
| | 15–56 | 55–75 | 113–23 | 78–45 | 83–7 | 87–75 | 39–52 | 61–13 | 64–92 |

**Table 6.** *Cont.*

| | Industrial | | | Residential | | | Commercial | | |
|---|---|---|---|---|---|---|---|---|---|
| | **Aver** | **Min** | **Max** | **Aver** | **Min** | **Max** | **Aver** | **Min** | **Max** |
| SHAMODE | 83–60 | 9–16 | 15–72 | 27–17 | 93–41 | 73–47 | 118–18 | 81–33 | 67–113 |
| | 46–108 | 68–76 | 64–5 | 86–88 | 3–57 | 65–11 | 7–6 | 2–98 | 94–41 |
| | 21–48 | 90–105 | 87–9 | 9–89 | 25–23 | 68–91 | 95–116 | 103–56 | 90–92 |
| | 34–90 | 15–52 | 26–82 | 53–16 | 5–13 | 14–112 | 24–97 | 15–69 | 29–23 |
| | 75–100 | 82–88 | 22–4 | 92–72 | 96–94 | 69–58 | 119–98 | 88–14 | 53–55 |
| | 24–114 | 73–104 | 90–17 | 117–55 | 40–19 | 100–53 | 77–64 | 53–65 | 75–104 |
| | 43–44 | 95–3 | 98–94 | 5–94 | 18–47 | 25–44 | 62–23 | 94–28 | 114–87 |
| | 12–66 | 91–67 | 67–100 | 22–60 | 84–85 | 2–79 | 33–44 | 84–116 | 117–86 |
| | 69–94 | 45–108 | 56–59 | 69–79 | 104–7 | 24–43 | 81–73 | 60–61 | 47–38 |
| | 7–58 | 46–42 | 80–71 | 41–118 | 79–100 | 86–46 | 11–39 | 91–52 | 98–91 |
| | 76–23 | 18–62 | 93–110 | 81–38 | 39–44 | 77–17 | 72–69 | 19–18 | 7–59 |
| | 53–72 | 43–5 | 31–73 | 44–64 | 89–81 | 48–29 | 61–91 | 105–62 | 10–51 |
| | 87–35 | 14–106 | 97–43 | 8–90 | 52–4 | 15–61 | 12–51 | 5–8 | 3–99 |
| | 98–22 | 13–2 | 46–11 | 52–6 | 48–27 | 56–52 | 106–56 | 13–23 | 46–77 |
| | 49–25 | 87–12 | 58–23 | 18–106 | 90–87 | 119–67 | 85–53 | 17–16 | 84–8 |
| MOFPA | 32–109 | 11–61 | 104–29 | 2–28 | 80–59 | 40–25 | 41–11 | 40–41 | 63–19 |
| | 91–60 | 102–107 | 95–116 | 72–78 | 39–114 | 7–17 | 48–102 | 14–61 | 2–58 |
| | 63–71 | 82–74 | 88–4 | 49–87 | 94–34 | 65–104 | 60–58 | 77–3 | 46–17 |
| | 76–54 | 84–14 | 7–102 | 32–12 | 87–63 | 23–12 | 43–87 | 33–88 | 60–65 |
| | 95–75 | 28–99 | 100–110 | 26–55 | 12–108 | 72–77 | 26–35 | 27–94 | 31–35 |
| | 26–99 | 96–58 | 73–8 | 31–83 | 95–37 | 31–110 | 65–68 | 52–84 | 43–101 |
| | 49–30 | 22–46 | 60–107 | 23–101 | 43–7 | 62–34 | 70–84 | 22–34 | 49–18 |
| | 59–21 | 79–26 | 75–117 | 67–43 | 16–5 | 118–112 | 30–103 | 106–26 | 68–54 |
| | 51–101 | 41–94 | 119–58 | 18–45 | 98–54 | 52–29 | 75–24 | 86–80 | 34–41 |
| | 25–79 | 89–111 | 113–56 | 99–38 | 62–105 | 14–32 | 56–100 | 60–59 | 3–75 |
| | 5–64 | 16–38 | 22–15 | 64–113 | 65–110 | 48–9 | 28–97 | 72–83 | 90–116 |
| | 73–35 | 86–18 | 52–111 | 119–79 | 25–58 | 11–42 | 115–64 | 20–108 | 115–72 |
| | 34–28 | 45–76 | 105–27 | 62–74 | 93–14 | 71–88 | 39–10 | 114–67 | 10–33 |
| | 114–100 | 33–29 | 31–97 | 110–71 | 32–117 | 36–99 | 77–104 | 97–6 | 109–118 |
| | 33–9 | 50–97 | 69–32 | 89–61 | 69–23 | 46–79 | 117–44 | 58–104 | 7–51 |
| MOGFPA | 92–38 | 35–64 | 81–2 | 89–106 | 103–21 | 50–100 | 99–112 | 106–40 | 60–13 |
| | 16–52 | 45–67 | 66–89 | 108–91 | 36–56 | 41–44 | 65–56 | 27–45 | 17–76 |
| | 24–5 | 100–105 | 75–35 | 16–56 | 34–51 | 74–106 | 105–94 | 20–67 | 11–21 |
| | 13–78 | 26–89 | 6–87 | 44–6 | 104–7 | 80–63 | 6–90 | 112–41 | 74–113 |
| | 56–90 | 101–39 | 25–108 | 35–5 | 33–60 | 56–16 | 95–8 | 49–52 | 6–40 |
| | 53–77 | 5–71 | 39–67 | 13–50 | 80–30 | 114–54 | 2–18 | 25–104 | 109–110 |
| | 88–82 | 34–94 | 118–13 | 34–43 | 22–16 | 110–72 | 38–76 | 90–113 | 28–95 |
| | 18–95 | 83–40 | 83–14 | 94–63 | 75–35 | 95–60 | 93–101 | 94–107 | 41–118 |
| | 19–86 | 7–25 | 32–41 | 39–14 | 8–44 | 94–42 | 82–45 | 62–3 | 18–87 |
| | 71–10 | 63–95 | 62–49 | 103–15 | 19–17 | 118–88 | 39–70 | 54–96 | 10–84 |
| | 20–109 | 77–103 | 69–59 | 84–68 | 94–31 | 23–76 | 89–22 | 11–65 | 52–34 |
| | 72–9 | 10–51 | 42–77 | 71–62 | 73–64 | 17–19 | 16–27 | 57–24 | 68–27 |
| | 112–93 | 48–85 | 80–53 | 98–51 | 45–23 | 89–58 | 11–42 | 77–66 | 98–55 |
| | 57–11 | 56–46 | 57–19 | 52–102 | 90–76 | 30–101 | 21–61 | 75–34 | 105–26 |
| | 54–33 | 119–47 | 113–103 | 12–19 | 58–42 | 98–21 | 47–109 | 9–110 | 78–8 |

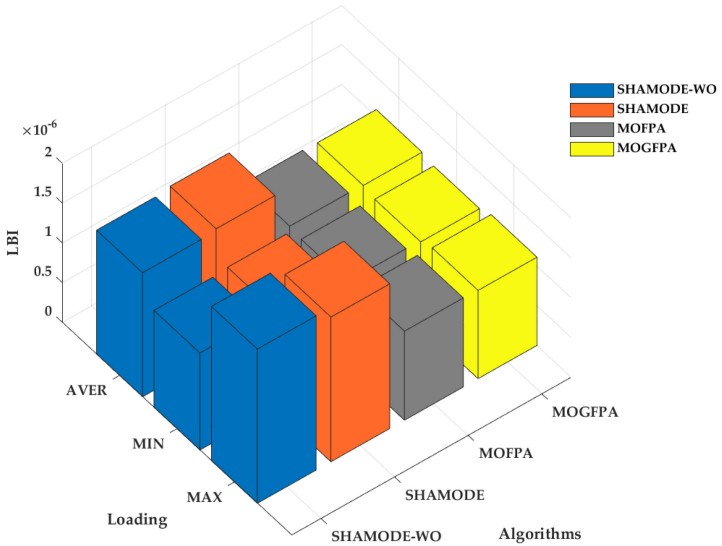

**Figure 4.** Comparative analysis of LBI for an industrial load on IEEE 119 bus system.

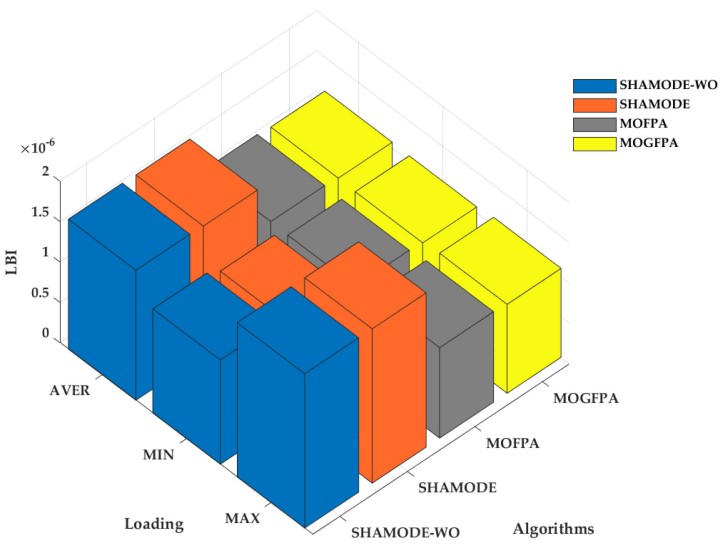

**Figure 5.** Comparative analysis of LBI for a residential load on IEEE 119 bus system.

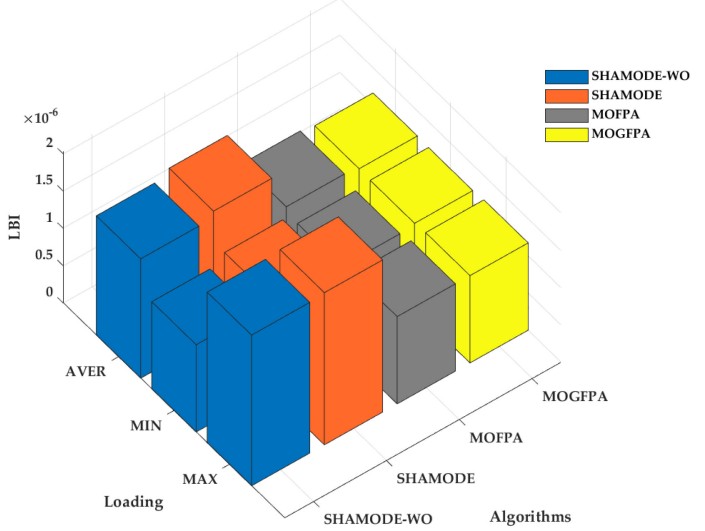

**Figure 6.** Comparative analysis of LBI for a commercial load on IEEE 119 bus system.

### 4.2. Evaluation of the Indian 52 Bus System

From a practical standpoint, the Indian 52 bus system of the smart urban system is used to evaluate performance in five instances. To meet the complete network requirement, the test system consists of 52 buses, 51 branches, and three feeders. The power factor of the test system is 0.9, which is a lag. The base MVA and kV for the desired test are 1 and 11, respectively. The bus voltage restrictions of the test system have magnitudes of 0.9 p.u and 1.05 p.u. The average, lowest, and maximum active PL for 52 bus realistic distribution networks with DG are compiled here. The proposed technique helps to lessen the bus system PLs while keeping the LBI and power balance within limits. Initially, the PLs in the system are discovered and mitigated through DG placement in a PS.

Furthermore, the desired method can find the best route for rerouting power flow and approach the problem as an MO numerical problem to seek optimal tie switch connections (Table 7) to lessen loss even further. This technique achieves outstanding results in decreasing PLs (Table 8) and when related to other previous approaches. Moreover, the comparative analysis among different methods under different load conditions is presented in Figure 7a–i. From all these figures, MOGFPA has yielded a fair Pareto optimal front containing all non-dominant solutions compared to MOFPA, SHAMODE, and SHAMODE–WO. The computation time of the MOGFPA is comparatively lesser than other algorithms. Henceforth, the MOGFPA is superior to all different algorithms.

**Table 7.** Tie switches configuration for the Indian 52 bus system.

|  | Aver | Min | Max | Aver | Min | Max | Aver | Min | Max |
|---|---|---|---|---|---|---|---|---|---|
| SHAMODE–WO | 29–10 | 20–31 | 49–40 | 25–47 | 42–40 | 37–31 | 30–15 | 28–20 | 42–52 |
|  | 14–38 | 6–34 | 28–52 | 18–29 | 52–28 | 10–17 | 13–40 | 21–44 | 28–51 |
|  | 40–42 | 8–33 | 31–19 | 42–28 | 46–51 | 36–52 | 24–51 | 19–12 | 23–34 |
|  | 34–45 | 11–18 | 33–46 | 16–20 | 13–47 | 26–16 | 23–42 | 31–18 | 15–12 |
|  | 18–5 | 10–40 | 34–11 | 37–10 | 14–25 | 25–11 | 46–31 | 29–4 | 47–10 |
|  | 9–11 | 13–23 | 41–36 | 11–31 | 11–48 | 29–51 | 25–16 | 13–52 | 14–5 |
| SHAMODE | 52–33 | 37–47 | 6–45 | 31–40 | 13–5 | 5–36 | 40–9 | 31–47 | 13–39 |
|  | 25–10 | 42–28 | 10–18 | 15–33 | 36–8 | 18–25 | 24–52 | 49–15 | 8–47 |
|  | 23–5 | 11–32 | 28–33 | 12–13 | 25–41 | 13–3 | 34–42 | 42–6 | 5–15 |
|  | 11–40 | 12–16 | 21–15 | 30–28 | 30–31 | 28–45 | 49–10 | 16–18 | 51–37 |
|  | 4–42 | 34–25 | 25–31 | 34–10 | 21–47 | 29–9 | 25–44 | 10–28 | 28–10 |
|  | 16–29 | 13–31 | 29–19 | 5–11 | 34–11 | 31–37 | 5–31 | 8–34 | 31–33 |
| MOFPA | 16–5 | 16–26 | 45–41 | 33–9 | 48–32 | 17–32 | 27–14 | 38–5 | 2–38 |
|  | 27–44 | 10–23 | 24–17 | 34–39 | 35–33 | 11–39 | 26–29 | 44–39 | 9–25 |
|  | 28–8 | 39–46 | 30–34 | 42–25 | 43–15 | 22–36 | 28–38 | 48–12 | 30–44 |
|  | 23–3 | 13–25 | 10–26 | 23–11 | 40–21 | 43–6 | 39–23 | 9–43 | 29–27 |
|  | 29–14 | 52–17 | 4–27 | 4–47 | 28–23 | 5–12 | 47–25 | 14–28 | 52–16 |
|  | 30–32 | 34–19 | 44–33 | 18–38 | 41–6 | 25–33 | 8–34 | 32–20 | 32–10 |
| MOGFPA | 5–22 | 24–28 | 39–37 | 43–25 | 34–49 | 17–5 | 20–48 | 15–37 | 17–5 |
|  | 35–52 | 47–32 | 52–50 | 46–33 | 37–40 | 20–40 | 10–6 | 42–39 | 20–40 |
|  | 34–2 | 19–22 | 26–14 | 11–48 | 48–21 | 34–4 | 23–49 | 19–8 | 34–4 |
|  | 18–23 | 38–23 | 23–25 | 35–45 | 36–31 | 42–29 | 38–2 | 29–4 | 42–29 |
|  | 43–46 | 6–27 | 2–3 | 22–37 | 10–5 | 43–30 | 4–3 | 45–20 | 43–30 |
|  | 19–50 | 12–14 | 31–43 | 42–38 | 39–15 | 37–28 | 16–42 | 32–49 | 37–28 |

**Table 8.** PL and LBI comparison under different load conditions for the Indian 52 bus system.

| | | | SHAMODE–WO | SHAMODE | MOFPA | MOGFPA |
|---|---|---|---|---|---|---|
| Industrial | Aver | PL(kW) | 560.18 | 559.67 | 456.06 | 450.67 |
| | | LBI | $1.56 \times 10^{-6}$ | $1.59 \times 10^{-6}$ | $1.15 \times 10^{-6}$ | $1.10 \times 10^{-6}$ |
| | Min | PL | 441.91 | 442.38 | 425.23 | 418.39 |
| | | LBI | $1.22 \times 10^{-6}$ | $1.24 \times 10^{-6}$ | $1.12 \times 10^{-6}$ | $1.07 \times 10^{-6}$ |
| | Max | PL | 675.43 | 675.67 | 488.89 | 470.58 |
| | | LBI | $1.94 \times 10^{-6}$ | $1.82 \times 10^{-6}$ | $1.18 \times 10^{-6}$ | $1.12 \times 10^{-6}$ |
| Residential | Aver | PL | 576.95 | 577.07 | 461.54 | 457.33 |
| | | LBI | $1.61 \times 10^{-6}$ | $1.60 \times 10^{-6}$ | $1.14 \times 10^{-6}$ | $1.10 \times 10^{-6}$ |
| | Min | PL | 445.36 | 445.64 | 420.09 | 418.54 |
| | | LBI | $1.29 \times 10^{-6}$ | $1.20 \times 10^{-6}$ | $1.18 \times 10^{-6}$ | $1.09 \times 10^{-6}$ |
| | Max | PL | 704.74 | 705.66 | 501.94 | 499.56 |
| | | LBI | $1.91 \times 10^{-6}$ | $1.91 \times 10^{-6}$ | $1.15 \times 10^{-6}$ | $1.12 \times 10^{-6}$ |
| Commercial | Aver | PL | 583.99 | 582.93 | 456.7 | 454.39 |
| | | LBI | $1.60 \times 10^{-6}$ | $1.69 \times 10^{-6}$ | $1.20 \times 10^{-6}$ | $1.16 \times 10^{-6}$ |
| | Min | PL | 450.75 | 446.02 | 418.21 | 410.29 |
| | | LBI | $1.17 \times 10^{-6}$ | $1.28 \times 10^{-6}$ | $1.17 \times 10^{-6}$ | $1.12 \times 10^{-6}$ |
| | Max | PL | 714.56 | 714.33 | 502.73 | 500.38 |
| | | LBI | $2.02 \times 10^{-6}$ | $2.04 \times 10^{-6}$ | $1.17 \times 10^{-6}$ | $1.15 \times 10^{-6}$ |

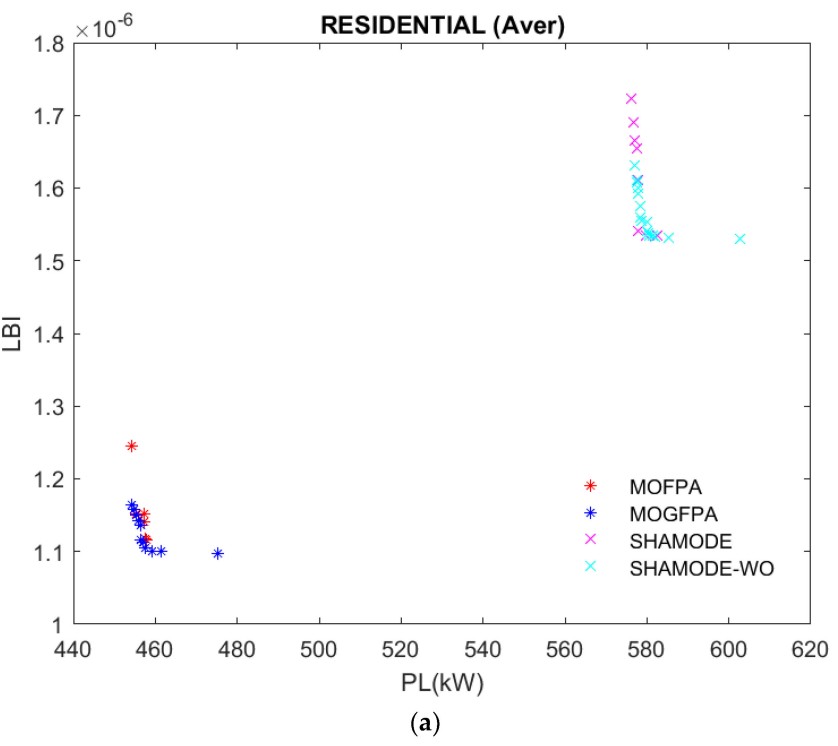

(**a**)

**Figure 7.** *Cont.*

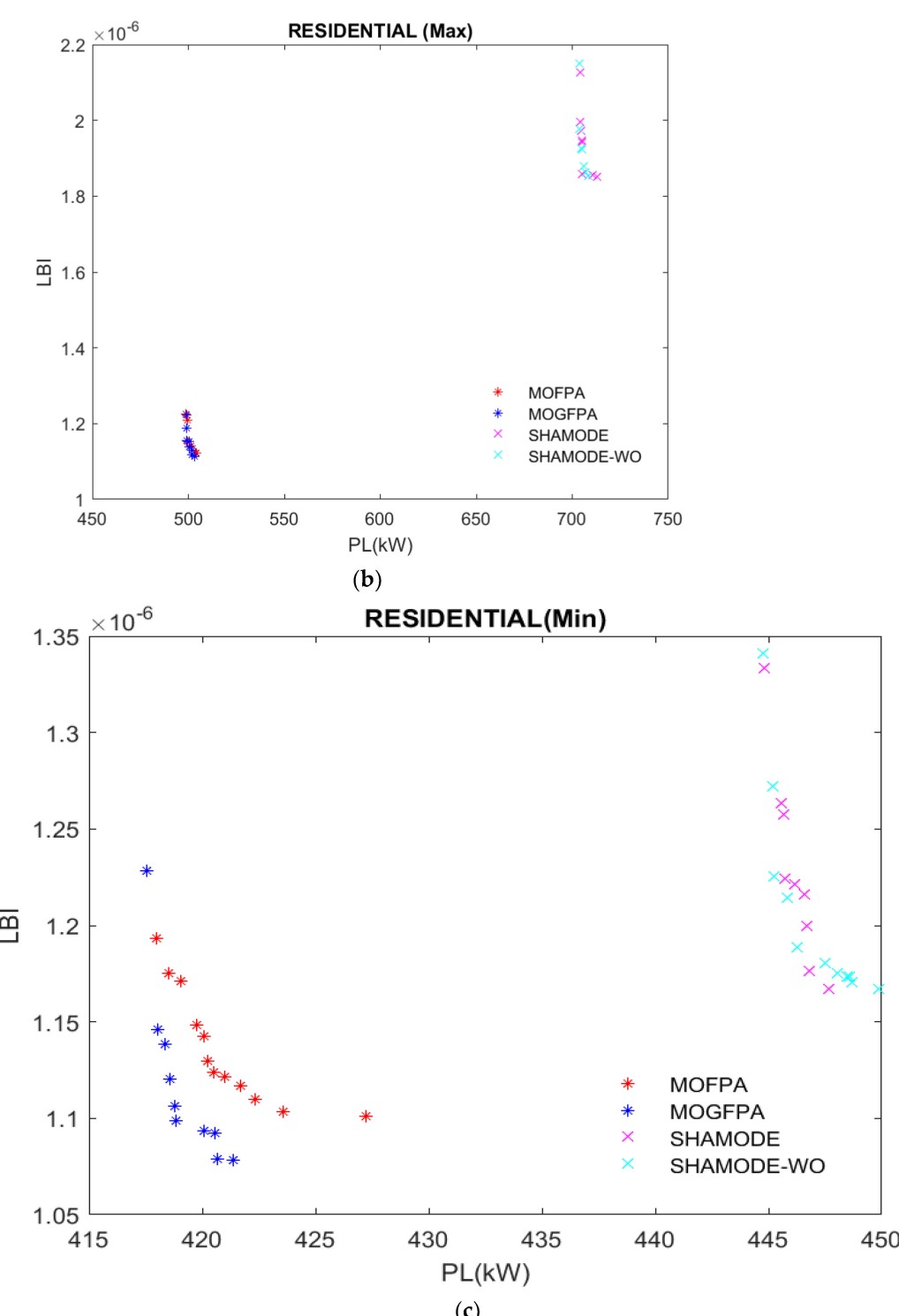

**Figure 7.** *Cont.*

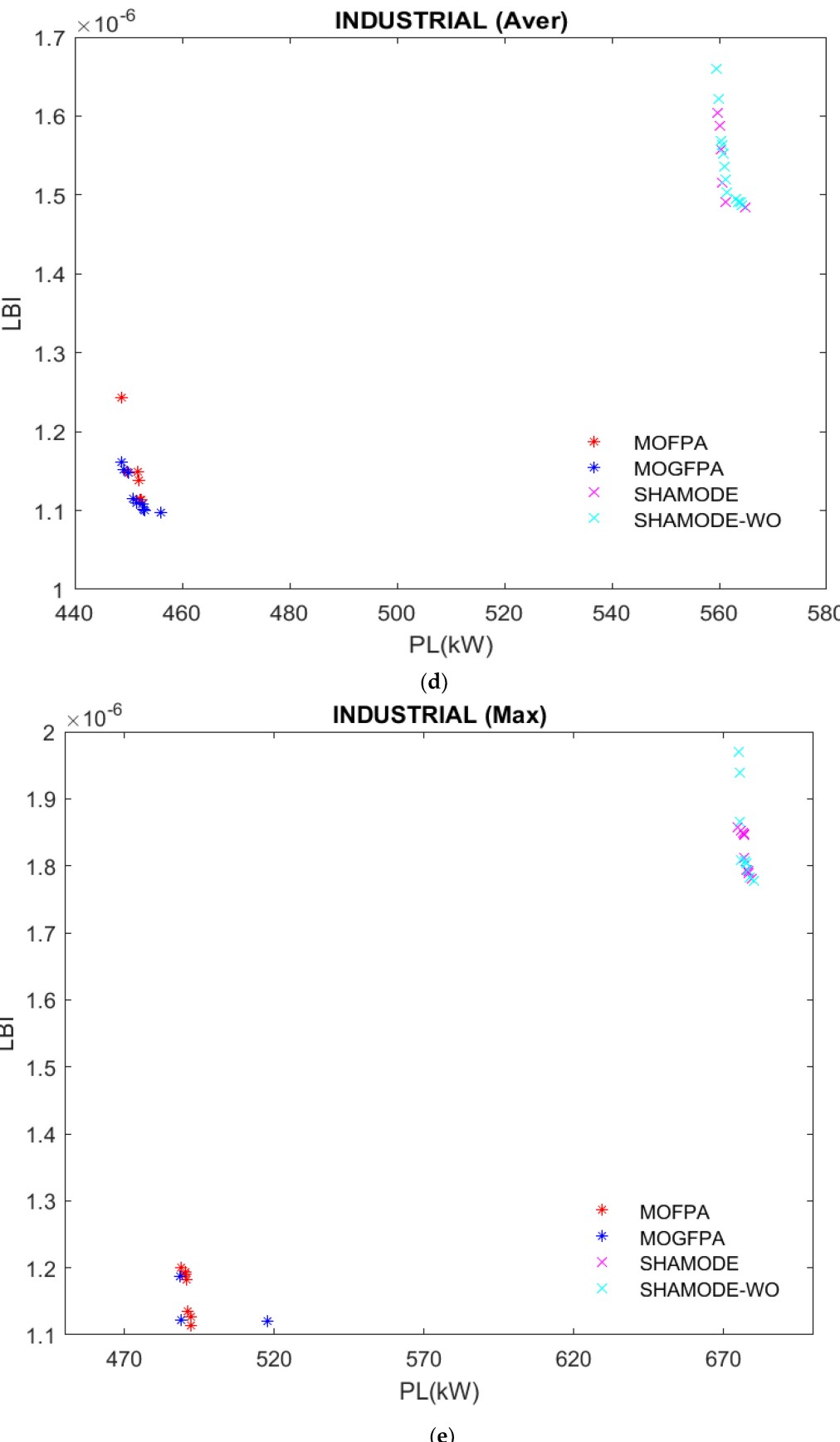

(**d**)

(**e**)

**Figure 7.** *Cont.*

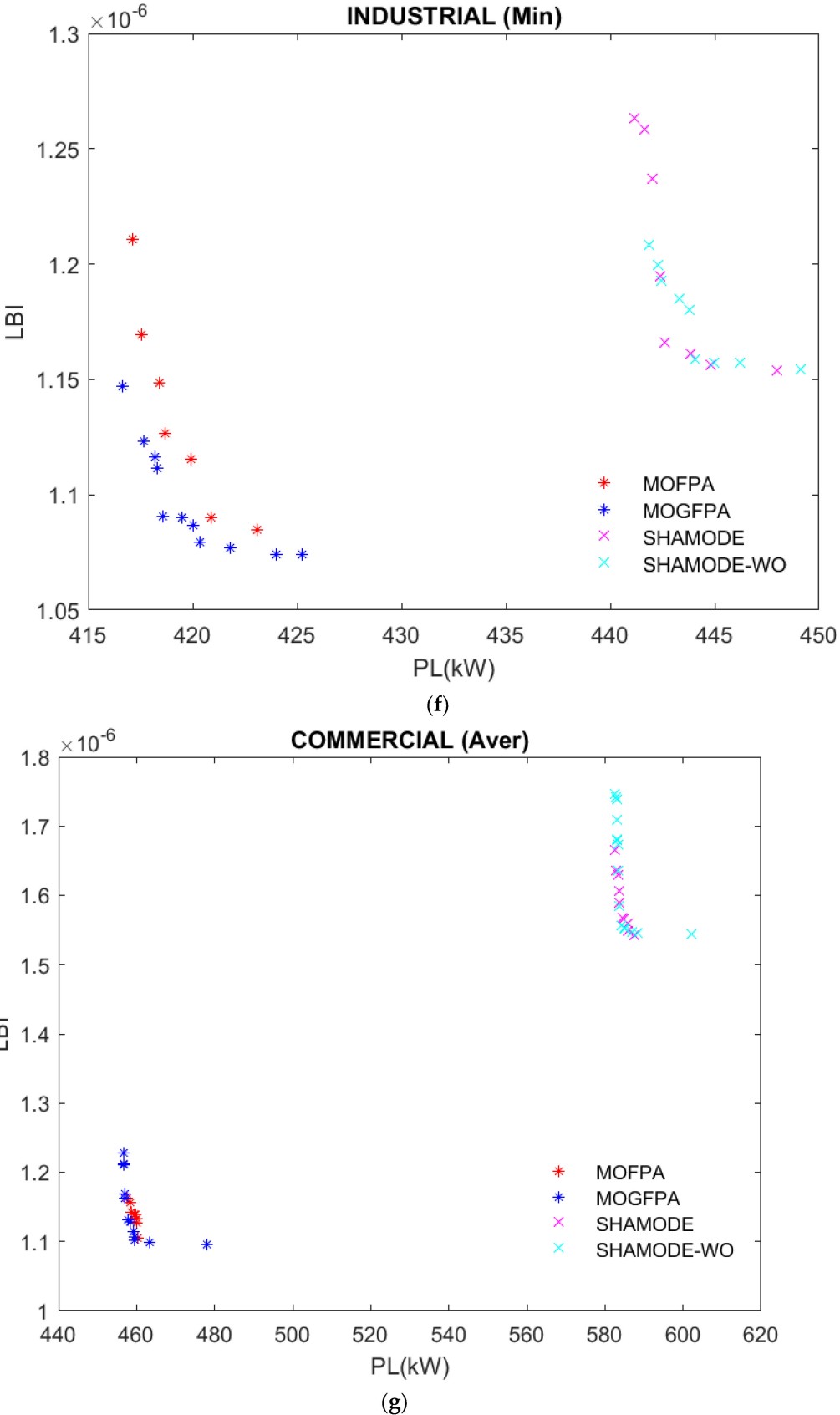

(f)

(g)

**Figure 7.** *Cont.*

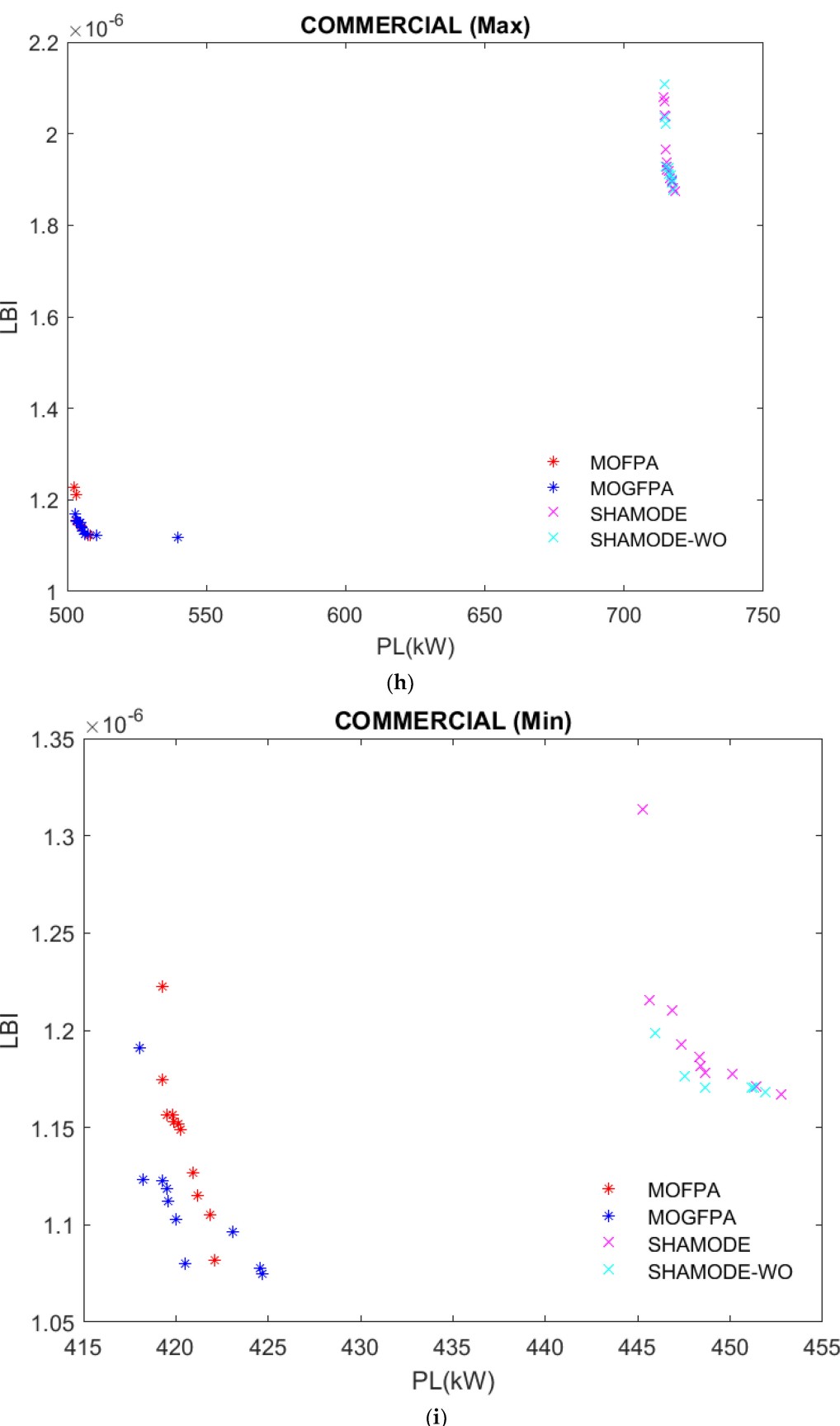

(**h**)

(**i**)

**Figure 7.** (**a**) Comparative analysis of average LBI and PLs for residential loads; (**b**) Comparative analysis for maximum LBI and PLs for residential loads; (**c**) Comparative analysis for minimum LBI

and PLs for residential loads; (**d**) Comparative analysis for average LBI and PLs for industrial loads; (**e**) Comparative analysis for maximum LBI and PLs for industrial loads; (**f**) Comparative analysis for minimum LBI and PLs for industrial loads; (**g**) Comparative analysis for average LBI and PLs for commercial loads; (**h**) Comparative analysis for maximum LBI and PLs for commercial loads; (**i**) Comparative analysis for minimum LBI and PLs for commercial loads.

## 5. Conclusions and Future Work

This paper formulated the optimal reconfiguration issue with an objective function as real power loss (PL) and load balance index (LBI) minimization. It makes the optimization algorithm a multi-objective (MO) problem that comprises multiple indices. The multi-objective golden flower pollination algorithm (MOGFPA) is stated to identify the ideal feeder connections in the RDS after properly allocating distributed generation (DG) to fulfil a target function. To demonstrate the proposed algorithm, its performance has been predictable with the Indian 52 bus and IEEE systems such as 33, 69, and 119 buses. Furthermore, the algorithm is tested under different load-varying conditions cases and compared with another state-of-the-art optimization algorithm. It was observed that the proposed novel MOGFPA provides the optimal global solution in the minimum period, which ultimately increases the load ability of the system and reduces PLs.

Future studies plan to integrate optimal energy storage system (ESS) allocation with reconfiguration studies.

**Author Contributions:** Conceptualization, D.S. and A.R.; methodology, D.S.; software, D.S.; validation, D.S. and A.R.; investigation, D.S.; data curation, writing—original draft preparation, D.S.; writing—review and editing, A.R.; visualization, D.S.; supervision, A.R. All authors have read and agreed to the published version of the manuscript.

**Funding:** This research received no external funding.

**Institutional Review Board Statement:** Not applicable.

**Informed Consent Statement:** Not applicable.

**Data Availability Statement:** Not applicable.

**Acknowledgments:** The authors are grateful to the authorities of Vellore Institute of Technology, Chennai, for progressively providing all facilities to perform this research work.

**Conflicts of Interest:** The authors declare no conflict of interest.

## Nomenclature

| | |
|---|---|
| ADG | Allocation of distributed generation |
| CABC | Chaotic artificial bee colony algorithm |
| CPSO | Chaotic particle swarm optimization |
| CSA | Clonal selection algorithm |
| CSO | Cuckoo search optimization |
| CSS | Centralized scheduling system |
| DG | Distributed generation |
| DN | Distribution network |
| DSM | Demand side management |
| EC | Energy consumption |
| ES | Evolution strategies |
| ESS | Energy storage system |
| EVs | Electric vehicles |
| FACTS | Flexible alternating current transmission system |
| FPA | Flower pollination algorithm |

| | |
|---|---|
| GA | Genetic algorithm |
| GP | Genetic programming |
| GS | Golden search |
| HEVs | Hybrid electric vehicles |
| HVDC | High-voltage direct current |
| LBI | Load balance index |
| MAs | Metaheuristic algorithms |
| MFPA | Modified flower pollination algorithm |
| MO | Multi-objective |
| MOFPA | Multi-objective flower pollination algorithm |
| MOGFPA | Multi-objective golden flower pollination algorithm |
| PDS | Power distribution system |
| PEV | Plug-in electric vehicle |
| PL | Power loss |
| PS | Power system |
| PSO | Particle swarm optimization |
| PV | Photovoltaics |
| RE | Renewable energy |
| RER | Renewable energy resources |
| SG | Smart grid |
| SHAMODE | Successive history adaptive multi-objective differential evolution |
| SHAMODE–WO | Successive history adaptive multi-objective differential evolution–whale optimization |
| **Notations** | |
| Aver | Average |
| Max | Maximum |
| Min | Minimum |
| $LoF$ | Load of factor |
| $pu\ demand$ | Per unit demand |
| $Pok(t)$ | Active power that is generated at bus '$k$' |
| $Qok(t)$ | Reactive power that is generated at bus '$k$' |
| $Powk(t)$ | At nominal voltage, bus '$k$' has an active load |
| $Qowk(t)$ | At nominal voltage, bus '$k$' has a reactive load |
| $Vk(t)$ | The voltage at bus $k$ |
| PDeF | Probability density beta function |
| $F(s)$ | Beta probability distribution for PV radiance |
| $\tau$ | Probability density function |
| $\alpha$ | Mean value |
| $\beta$ | Standard deviation |
| $So$ | Random variable |
| $O.F1$ | Objective function 1 |
| $O.F2$ | Objective function 2 |
| $P_{loss}$ | Active power loss |
| $I_j$ | Branch current |
| $R_j$ | Line resistance |
| $I$ | Rated current of the bus |
| $V_{min}$, $V_{max}$ | Minimum and maximum levels of voltage |
| $V_i$ | The voltage at bus $i$ |
| $N^{BR}$ | Number of branches |
| $I_{max,j}$ | Maximum flow of current in PDS |
| $P_s$ | Power at slack bus |
| $N_G$ | Number of generator buses |
| $P_d$ | Power demand at the load end |
| $P(x)$ | Pareto probability distribution |
| $b$ | Scale parameter |
| $a$ | Shape parameter |
| D | Size of an optimization problem |

| | |
|---|---|
| $d$ | Direction of flight |
| $x$ | Fitness parameter |
| $g*$ | Global best solution |
| $stp$ | Step size |
| $X_t$ | Solution vector at iteration t |
| $X_{t+1}$ | Solution vector at iteration t+1 |
| $\Phi$ | Golden ratio |
| $\alpha$ | The inverse of the golden ratio |

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
