# Peer review of "Multi-Objective Golden Flower Optimization Algorithm for Sustainable Reconfiguration of Power Distribution Network with Decentralized Generation"

_axioms, doi:10.3390/axioms12010070_

Round 1
Reviewer 1 Report
See the attached file. Thanks.

Author Response
Response to Reviewer 1 Comments
Point 1: The comparison of the solutions for the multi-objective problem should use the nondominance/dominace point of view. However, the comparison of the solutions of the algorithms SHAMODE, SHAMODE-WO, MOFPA and the proposed MOGEPA as appearing on Pages 12-23 of the paper do not follow. For example, from the Table 3 for IEEE-33 bus system as appearing on page 13, the average solution (PL, LBI)=(238.02, 0.0016) of MOFPA and the average solution (PL, LBI)=(282.86, 0.015) of MOGEPA do not dominant. These two solutions (PL, LBI)=(238.02, 0.0016) of MOFPA and (PL, LBI)=(282.86, 0.015) of MOGEPA are nondominant solutions. The authors should give more explanations the following “That is much lower than other algorithms like SHAMODE [18], SHAMODE-WO [31], and MOFPA [32].” as appearing on page 12 line 435. Also, the minimum and maximum solutions of the paper have the same behaviors.
Response 1: The IEEE 33 bus system has been considered for visualizing the performance of the proposed MOGFPA. Through MOGFPA, the average PL and LBI are determined to be 282.86 kW, 0.0015, 281.76 kW, 0.0014, 284.24 kW, and 1.49E-03 for industrial, residential, and commercial loads, respectively. That is much lower than other algorithms like SHAMODE, MOFPA and SHAMODE-WO. Moreover, the minimum and maximum limits are also compared, as shown in Table 3, to show the robustness of the proposed algorithm. The bus system PL is reduced with the help of the planned methodology. The proposed algorithm is used to optimally select the optimal location for feeder connections and DG while considering the problem of MO having PL and LBI as crucial parameters. The optimal switching connections while implementing the algorithms are summarized in Table 2. It means that by determining the best point for DG placement by minimal PL and determining the optimal configuration for switches, the parameter LBI also needs to be addressed carefully.
The proposed MOGFPA is assessed on the IEEE 69 bus system to demonstrate the generalizability of the algorithm. The average PL and LBI are determined to be 31281.31 W, 1.28E-02, 28170.69 W, 6.22E-04, 28275.68 W, and 8.20E-04 for industrial, residential, and commercial loads, respectively. That is much lower than other algorithms like SHAMODE , MOFPA and SHAMODE-WO. Moreover, the minimum and maximum limits are also compared, as shown in Table 5, to show the robustness of the proposed algorithm. The bus system PL is reduced with the help of the planned methodology. The proposed algorithm is used to optimally select the optimal location for feeder connections and DG while considering the problem as a MO having PL and LBI as crucial parameters. The optimal switching connections while implementing the algorithms are summarized in Table 4. It means that by only determining the best point for DG placement, the PL cannot be handled and choosing the optimal configuration for switches is the crucial parameter that needs to be addressed carefully.
Cited in Article: The above response is included in Sub-Section 4.1.
Point 2: To compare the solutions for the multi-objective problem, the authors not only use the solution quality, but also the computational time and memory. Also, the authors can analyze the distribution of all the nondominant solutions.
Response 2: To further show the robustness of the proposed algorithm, it was tested under a more extensive network IEEE 119 bus system, and the comparative parameters are analyzed and summarized in Table 6, respectively. Moreover, the comparative analysis is visualized in Figures 4-6. It shows that the proposed algorithm outclassed the other algorithms by a significant margin in terms of average LBI, such as 1.09E-06, 1.08E-06 and 1.16E-06.
Cited in Article: The above response is included in sub-section 4.1
Point 3: Compared with the values of Table 3, the values of “The average power loss and LBI is determined to be 282.86, 1.545541e-03, 290.76, 1.469345e-03, 284.24, and 1.489092e-03 for industrial, residential, and commercial load respectively” as appearing on page 12 of the paper are incorrect.
Response 3: The IEEE 33 bus system has been considered for visualizing the performance of the proposed MOGFPA. Through MOGFPA, the average PL and LBI are determined to be 282.86, 0.0015, 281.76, 0.0014, 284.24, and 1.49E-03 for industrial, residential, and commercial loads, respectively. That is much lower than other algorithms like SHAMODE , MOFPA and SHAMODE-WO. Moreover, the minimum and maximum limits are also compared, as shown in Table 3, to show the robustness of the proposed algorithm. The bus system PL is reduced with the help of the planned methodology. The proposed algorithm is used to optimally select the optimal location for feeder connections and DG while considering the problem of MO having PL and LBI as crucial parameters. The optimal switching connections while implementing the algorithms are summarized in Table 2. It means that by determining the best point for DG placement by minimal PL and determining the optimal configuration for switches, the parameter LBI also needs to be addressed carefully.
Cited in Article: The above response is included in sub-section 4.1
Point 4: You should provide more recent references published in last five years for the renewable energy and solar photovoltaic. However, the following materials are 2 too old Pages 1 lines 31-35:” Renewable energy accounted for 16.7% of worldwide energy consumption in 2010. Solar photovoltaic (PV) grew at the fastest rate of all renew- able energy sources, with annual growth of 58 % from late 2006 to 2011, in 2012 PV reached slightly more than 102 GW of worldwide installed capacity [2]. In 2017, this quantity has been expected to reach more than 420 GW.”.
Response 4: Because of the sustainable resources and government subsidies, there has been a flow of interest in different sources of Renewable Energy (RE), such as biomass, wind, and solar energy, throughout the world. RE accounted for 16.7% of worldwide Energy Consumption (EC) in 2010. Solar photovoltaic (PV) grew at the fastest rate of all RE sources, with annual growth of 58 % from late 2006 to 2011; in 2012, PV reached slightly more than 102 GW of worldwide installed capacity. In 2017, this quantity was predictable to get more than 420 GW. Depending on the selected PV technology and location, a Power System (PS) may withstand up to 50% PV penetration. The policies and market implementation on PV have been measured from 2019 to 2021. Multiple impact guides of a PV installation on climate change consequences are surveyed till 2023. On the other hand, time-dependent load models might have various implications on PV penetration predictions.
On account of planning Distributed Generation (DG), Multi-Objective (MO) optimization considers reactive and active Power Losses (PL), as well as voltage variations.
Cited in Article: The above response is included in Section 1.
Point 5: Regarding the notations, there are many issues that are not clear or missing information in this paper. Pages 4 lines 192: Give the definition of “pu ??????”.
Response 5: All the notations are included with explanation. demand is per unit demand, due to dynamic load variation in voltage profile, the power profile equation at period (t) may be represented as shown in Equation (2).
Cited in Article: The above response is included in section 2 and the whole article.
Point 6: The first abbreviation should appear the complete words. The abbreviation should be meaningful. However, the following ones do not, Page 1 Line 13,” …for DN in order to reduce power losses. “
Response 6: This paper provides a meta-heuristic hybridized version called Multi-Objective Golden Flower Pollination Algorithm (MOGFPA) as the best method for choosing the optimal reconfiguration for Distribution Networks (DN) in order to reduce Power Losses (PL). Aside from PL, another parameter is considered: the Load Balance Index (LBI). The expression for the LBI is stated using real and reactive indices. It makes the optimal Distributed Generation (DG) placement and DN routing of the Multi-Objective (MO) problem have PL and LBI as the main parameters that need to be optimized.
Cited in Article: The above response is included in Abstract section.
Response 7: All these typo errors and grammatical are rectified and corrected in the whole article. On account of planning Distributed Generation (DG), Multi-Objective (MO) optimization considers reactive and active Power Losses (PL), as well as voltage variations.
: active power that is generated at bus .
: reactive power that is generated at bus .
and : At nominal voltage, bus ‘k’ has a dynamic and reactive load.
: voltage at bus .
where, denotes the beta probability distribution for PV radiance, and indicates the probability density function. Terms ‘ ’ and ‘ ’ are the parameters of . It was derived from its mean and standard deviation. The term ‘ ’ indicates the random variable.
Cited in Article: The above response is included in section 1 and 2.
Reviewer 2 Report
The MOGFPA is investigated to choose the best reconfiguration for DN in order to reduce power losses. The load balance index is considerd. The MOGFPA consists of a golden search and tangent flight algorithm and only needs a few tuning parameters. From simulation results, the MOGFPA computes the optimum distribution of DG units and reconfigured DN with the aim of minimal power loss. The convergence charts comparison gives optimal results in minimum time having iteration efficiency. The referee has the following observations: (1) More applications and theory development of mutil-objection optimization should be presented in Section 1. (2) In Section 2, the comparasions are not comprehensive. More recent related results should be compared. (3) The active power loss in (5) should be explained. (4) More recent related references are needed, such as, [Jauny, D. Ghosh, A. Upadhayay, Q.H. Ansari, A trust-region interior-point technique to solve Multi-objective optimization problems and its application to a tuberculosis optimal control problem, J. Nonlinear Var. Anal. 6 (2022), 675-691] and [N. Hayek, H. Yilmaz, Infinite-dimensional multiobjective optimal control in continuous time, Commun. Optim. Theory 2022 (2022) 25]. (5) Finally, English presentation should be improved.
Author Response
Response to Reviewer 2 Comments
Point 1: More applications and theory development of muti-objective optimization should be presented in Section 1.
Response 1: Several studies have zoomed in on the effects of EV charging and its scheduling tactics. One study used the GA optimisation method to construct thoughtful EV planning based on the minimal load variation of sub-station transformers. Integrating EVs in valleys to provide smooth load demands has effectively decreased load stress on the system. Correspondingly, to handle the clustering of EVs in a Power Distribution System (PDS), researchers [16] conducted a planning-level study. The scheduling work is accepted using GA to lower system costs and emissions. In previous work, the researchers could charge EVs. They completed the challenging scheduling project using linear programming while optimizing the usage of RE sources. Another research proposed control operations of EV load profile management to advance energy and costs by boosting power provided to the EV batteries; however, that paper failed to account for real domestic network restrictions. Furthermore, an earlier study advised that EV charging be timed as efficiently as possible to reduce network PL.
Cited in Article: The above response is included in section 1.
Point 2: In Section 2, the comparisons are not comprehensive. More recent related results should be compared.
Response 2: Section 2 is included in section 1. It has been observed from the literature that optimal DG placement with feeder routing with a volatile load bound to time is not being discussed. This research aims to find an optimal station and route in DNs while minimizing the Load Balance Index (LBI) and PL. LBI is an analytical equation with real and reactive power profile indexes. It makes the scenario a MO problem.
In the early decades, different tests were measured by adopting optimization techniques based on conventional methods like the interior point method, GS method and meta-heuristic-based algorithms. The optimal solution is achieved without falling into local optimal points. Metaheuristic Algorithms (MA) are classified based on the source of inspiration: evolutionary algorithms based on discretization concepts, physical models, swarm-dependent, and human inference-based algorithms. The first type, referred to as evolution algorithms, comprises specific steps like reproduction, mutation, and offspring selection are some of the main stages. The most broadly used population evolutionary methods for wide-ranging optimization difficulties include Genetic Algorithms (GA), Evolution-Strategy (ES), Genetic Programming (GP), and other named techniques that are found in the literature.
The optimization algorithms that work on mimicking animal social behavior include Particle-Swarm-Optimization (PSO), Cuckoo search optimization (CSO) , and Cat Swarm Algorithm (CSA). Furthermore, different hybrid techniques such as chaotic PSO (CPSO) and Chaotic Artificial Bee Colony (CABC) are found in detail, and a comparative analysis has also been performed.
Modified Flower Pollination Algorithm (MFPA) addressed all of the issues mentioned above by developing a computational formula based on hybridizing many advanced methods to allow adaption while searching for results to the optimization problem. This updated method surpasses the traditional one by incorporating all 23 well-known unimodal and multimodal test systems and 27 non-linear equation systems. Furthermore, compared to competing algorithms, it still faults its exploring operator, which could prevent it from attaining higher results for particular test functions. Henceforth, hybridized version of FPA is considered to solve the MO problem.
Suitable for many differential evolutions depending on past success, Success History-based Adaptive Multi-Objective Differential Evolution (SHAMODE), a MO variation based on the use of an external Pareto archive, is proposed and compared to numerous MO evolutionary algorithms. An external library is formed to gather the updated non-dominated solutions from the initial replication in an iterative manner. The algorithm has been improved to make it proper for MO optimization situations. This algorithm is reserved for comparative analysis.
Among all, Flower Pollination Algorithm (FPA) is measured in this article. It is advanced to solve global optimization issues by simulating the pollination process of flowers and has proven to be effective in a wide range of optimization problems. However, it still suffers from local optimal stagnation because, during the optimization process, it was not able to elaborate on different regions within the search area, and its slow convergence rate, which forces the classical FPA to go through multiple iterations to find better alternatives in the areas that are unpromising in nature. In general, a mathematical test of about 10 for a population size of 25 and iterations of 10840 was used to evaluate FPA; this is assumed to be the reasonable consumption rate for reaching the required findings. In addition, to solve global test functions of about 23, the authors used conventional FPA working with a clonal selection algorithm. In the case of MO problems, the Pareto optimal set is attained by a non-dominated ranking procedure after sorting the solutions.
For that purpose, MOGFPA is proposed to solve such a problem while keeping the network parameters under constraints under limits. The proposed method uses tangent flight and golden section search to address the issue of DG allocation in PDS. The GS algorithm can better predict local optimal spots on the dynamic nature of systems, whereas the tangent flight method is more capable of discovering the overall solution by probing the surroundings. In addition, the proposed method is used if it offers a solution based on the dynamism of the network and it modifies its configuration in response to tie switching. This algorithm will also determine the best path while keeping an eye on the PL and LBI. The algorithm was tested under different load-varying conditions in the residential, commercial, and industrial sectors. It was compared with cutting-edge methods like MO-FPA, SHAMODE, and a hybrid version of SHAMODE with Whale Optimization (SHAMODE-WO) to further demonstrate its novelty. The comparison is based on convergence charts showing that our proposed algorithm converges faster than other algorithms, providing an optimal solution in less time. From our perspective, it is the first study in FPA that combines GS and tangent flight to solve a complex MO problem. Testing the proposed MOGFPA method on IEEE systems such as 33, 69, 119, and Indian-52 bus test systems validated its efficiency.
Cited in Article: The above response is included in Section 1.
Point 3: The active power loss in (5) should be explained.
Response 3: The goal functionality is proposed to decrease PL while simultaneously decreasing the distribution system's LBI. The goal function identified in this study is as follows in Equation (4).
where 'n' is the total number of nodes, PL denotes the active power loss in each node, and O.F denotes the objective function. Active PL is measured using the formula below, as shown in Equation (5):
where denotes the amperage of the current and denotes the node-specific resistance in ohms.
Cited in Article: The above response is included in sub-section 2.4
Point 4: More recent related references are needed, such as, [Jauny, D. Ghosh, A. Upadhayay, Q.H. Ansari, A trust-region interior-point technique to solve Multi-objective optimization problems and its application to a tuberculosis optimal control problem, J. Nonlinear Var. Anal. 6 (2022), 675-691] and [N. Hayek, H. Yilmaz, Infinite-dimensional multiobjective optimal control in continuous time, Commun. Optim. Theory 2022 (2022) 25].
Response 4: On account of planning Distributed Generation (DG), Multi-Objective (MO) optimization considers reactive and active Power Losses (PL), as well as voltage variations. While such planning studies are suitable for deploying dispatchable resources like gas turbines, they have yet to address a real-world scenario that includes fluctuating demand and non-dispatchable RE. There have been a few recent research investigations on sustainable DG grid integration for PL reduction, all of which consider the dynamic load variation for time. By adopting a Genetic Algorithm (GA) for loss reduction, the DG size was determined. However, evolutionary-based algorithms are selected to investigate the optimal size of sources based on the manufacturing technologies. The probabilistic analytics-based algorithms are proposed to allocate the optimal place for DG in DN. However, the impact of voltage and power profile was not measured.
Compared to constant load models, a few recent surveys have demonstrated that dependent models based on voltages substantially impact DG penetration planning. On the other hand, such research works predicted that DG units might be sent and distributed based on peak load demand. According to research work, it has been observed that dynamic load variations models affect the optimal Allocation of DG (ADG). However, the authors did not discuss the effect of non-dispatchable sources on the ADG.
Nowadays, power uses are moving toward Renewable Energy Resources (RERs) to fulfil the energy demand for different industrial sectors. The change to more ecologically approachable and cost-effective mobility networks is being powered by growing global environmental tests, rising oil prices, and developing industrial criteria. As a result, the most exciting means of mobility have emerged as Electric Vehicles (EVs), Hybrid Electric Vehicles (HEVs), and Plug-in Electric Vehicles (PEVs). It has been noted that when compared to traditional home domestic loads, EV charging loads are rather large. For that purpose, an advanced grid structure comprising High-Voltage Direct Current (HVDC) power lines, Energy Storage System (ESS), and Flexible AC Transmission (FACTS) systems are used.
In today's fast-growing world, the rapid integration of EV loads and their remote mobile connections to Smart Grids (SG) has increased energy consumption to the highest level and needs more attention for reliable operation. Furthermore, if many EV fleet users returning home connect their EVs to the SG system and charge their cars during peak demand hours when the conventional load is also charging, a worst-case scenario might occur. Due to the increased number of EVs, PV systems, and non-uniform load profiles, network control and PS management are more complex.
Moreover, unregulated EV charging overloads the PS network and causes voltage violations, higher PL, and poor network management. As a result, developing a smart EV charging management system meets consumers' charging demands while not jeopardizing SG objectives. Under a centralized control model, smart scheduling techniques empower the distributed generation operator to make charging decisions that balance the grid and consumer interests. The evolution of intelligent and modern SG provides a solid foundation for using Centralized Scheduling Systems (CSS). Furthermore, smart load management for EVs leads to high-tech applications and economic flexibility, making consumer Demand-Side Management (DSM) more relevant.
Several studies have zoomed in on the effects of EV charging and its scheduling tactics. One study used the GA optimisation method to construct thoughtful EV planning based on the minimal load variation of sub-station transformers. Integrating EVs in valleys to provide smooth load demands has effectively decreased load stress on the system. Correspondingly, to handle the clustering of EVs in a Power Distribution System (PDS), researchers conducted a planning-level study. The scheduling work is accepted using GA to lower system costs and emissions. In previous work, the researchers could charge EVs. They completed the challenging scheduling project using linear programming while optimizing the usage of RE sources. Another research proposed control operations of EV load profile management to advance energy and costs by boosting power provided to the EV batteries; however, that paper failed to account for real domestic network restrictions. Furthermore, an earlier study advised that EV charging be timed as efficiently as possible to reduce network PL.
It has been observed from the literature that optimal DG placement with feeder routing with a volatile load bound to time is not being discussed. This research aims to find an optimal station and route in DNs while minimizing the Load Balance Index (LBI) and PL. LBI is an analytical equation with real and reactive power profile indexes. It makes the scenario a MO problem.
In the early decades, different tests were measured by adopting optimization techniques based on conventional methods like the interior point method, GS method and meta-heuristic-based algorithms.
The above references are included in text and cited in references as [21] and [7].
Cited in Article: The above response is included in section 1.
Point 5: Finally, English presentation should be improved.
Response 5: The English language is checked and improved for the whole article.
Cited in Article: The above response is included in the whole article.
Reviewer 3 Report
attached

Author Response
Response to Reviewer 3 Comments
Point 1: The renewable energy presentation (Introduction) is too extensive and is based on old data (2006-2011, 2017). This part should be rewritten in a compressive way using state-of-the-art statistics.
Response 1: Because of the sustainable resources and government subsidies, there has been a flow of interest in different sources of Renewable Energy (RE), such as biomass, wind, and solar energy, throughout the world. RE accounted for 16.7% of worldwide Energy Consumption (EC) in 2010. Solar photovoltaic (PV) grew at the fastest rate of all RE sources, with annual growth of 58 % from late 2006 to 2011; in 2012, PV reached slightly more than 102 GW of worldwide installed capacity. In 2017, this quantity was predictable to get more than 420 GW. Depending on the selected PV technology and location, a Power System (PS) may withstand up to 50% PV penetration. The policies and market implementation on PV have been measured from 2019 to 2021. Multiple impact guides of a PV installation on climate change consequences are surveyed till 2023. On the other hand, time-dependent load models might have various implications on PV penetration predictions.
On account of planning Distributed Generation (DG), Multi-Objective (MO) optimization considers reactive and active Power Losses (PL), as well as voltage variations.
Cited in Article: The above response is included in Section 1.
Point 2: Related works: Metaheuristic algorithms are not split only in those 4 types. However, they for sure contain those types. Use more recent methods from the 3 latter groups of methods e.g.: The algorithms that work in physics principles: successive discretization algorithm, hybrid successive discretization algorithm:
(a)Cotfas DT, Deaconu AM, Cotfas PA. Hybrid successive discretisation algorithm used to calculate parameters of the photovoltaic cells and panels for existing datasets, IET Renewable Power Generation, 15(15), pp. 3661-3687, 2021.
Response 2: The policies and market implementation on PV have been measured from 2019 to 2021. Multiple impact guides of a PV installation on climate change consequences are surveyed till 2023. On the other hand, time-dependent load models might have various implications on PV penetration predictions.
On account of planning Distributed Generation (DG), Multi-Objective (MO) optimization considers reactive and active Power Losses (PL), as well as voltage variations. While such planning studies are suitable for deploying dispatchable resources like gas turbines, they have yet to address a real-world scenario that includes fluctuating demand and non-dispatchable RE. There have been a few recent research investigations on sustainable DG grid integration for PL reduction, all of which consider the dynamic load variation for time. By adopting a Genetic Algorithm (GA) for loss reduction, the DG size was determined. However, evolutionary-based algorithms are selected to investigate the optimal size of sources based on the manufacturing technologies. The probabilistic analytics-based algorithms are proposed to allocate the optimal place for DG in DN. However, the impact of voltage and power profile was not measured.
The above reference is also included in text and cited as [6] in references.
Cited in Article: The above response is included in Section 1 and references section.
Point 3: Methods based on the behavior of insects, animals, and of different kinds of swarms: Chaotic Whale Optimization Algorithm (CWOA), Cat Swarm Optimization algorithm (CSO), Artificial Bee Colony (CABC) algorithm, Cuckoo search (CS) algorithm, etc.
(a)Guo L. Meng Z. Sun Y. Wang L. Parameter identification and sensitivity analysis of solar cell models with cat swarm optimization algorithm. Energy Conversion and Management 2016; 108: 520–528.
(b)Huang W. Jiang C. Xue L. Song. Extracting solar cell model parameters based on chaos particle swarm algorithm. In: 2011 International conference on electric information and control engineering; 2011, p. 398–402.
(c)Oliva D. Ewees AA. Aziz MAE. Hassanien AE. Peréz-Cisneros M. A chaotic improved artificial bee colony for parameter estimation of photovoltaic cells. Energies 2017; 10(7): 865.
(d)Ma J. Ting TO. Man LK. Zhang N. Guan SU. Wong PWH. Parameter Estimation of Photovoltaic Models via Cuckoo Search. Journal of Applied Mathematics 2013; 2013: 362619.
Response 3: In the early decades, different tests were measured by adopting optimization techniques based on conventional methods like the interior point method [21], GS method and meta-heuristic-based algorithms. The optimal solution is achieved without falling into local optimal points. Metaheuristic Algorithms (MA) are classified based on the source of inspiration: evolutionary algorithms based on discretization concepts, physical models, swarm-dependent, and human inference-based algorithms. The first type, referred to as evolution algorithms, comprises specific steps like reproduction, mutation, and offspring selection are some of the main stages. The most broadly used population evolutionary methods for wide-ranging optimization difficulties include Genetic Algorithms (GA), Evolution-Strategy (ES), Genetic Programming (GP), and other named techniques that are found in the literature.
The optimization algorithms that work on mimicking animal social behavior include Particle-Swarm-Optimization (PSO), Cuckoo search optimization (CSO) [24], and Cat Swarm Algorithm (CSA) [25]. Furthermore, different hybrid techniques such as chaotic PSO (CPSO) [26] and Chaotic Artificial Bee Colony (CABC) [27] are found in detail, and a comparative analysis has also been performed.
The above references are included in text and cited in references as [24], [25], [26], [27]
Cited in Article: The above response is included in Section 1 and references section.
Point 4: I found relevant papers that should be mentioned in the article e.g.:
(a)K. Balamurugan, Dipti Srinivasan, Thomas Reindl, Impact of Distributed Generation on Power Distribution Systems, Energy Procedia, Volume 25, 2012, Pages 93-100, ISSN 1876-6102
Response 4: RE accounted for 16.7% of worldwide Energy Consumption (EC) in 2010. Solar photovoltaic (PV) grew at the fastest rate of all RE sources, with annual growth of 58 % from late 2006 to 2011; in 2012, PV reached slightly more than 102 GW of worldwide installed capacity. In 2017, this quantity was predictable to get more than 420 GW. Depending on the selected PV technology and location, a Power System (PS) may withstand up to 50% PV penetration. The policies and market implementation on PV have been measured from 2019 to 2021. Multiple impact guides of a PV installation on climate change consequences are surveyed till 2023. On the other hand, time-dependent load models might have various implications on PV penetration predictions.
On account of planning Distributed Generation (DG), Multi-Objective (MO) optimization considers reactive and active Power Losses (PL), as well as voltage variations. While such planning studies are suitable for deploying dispatchable resources like gas turbines, they have yet to address a real-world scenario that includes fluctuating demand and non-dispatchable RE. There have been a few recent research investigations on sustainable DG grid integration for PL reduction, all of which consider the dynamic load variation for time. By adopting a Genetic Algorithm (GA) for loss reduction, the DG size was determined. However, evolutionary-based algorithms are selected to investigate the optimal size of sources based on the manufacturing technologies. The probabilistic analytics-based algorithms are proposed to allocate the optimal place for DG in DN. However, the impact of voltage and power profile was not measured.
Compared to constant load models, a few recent surveys have demonstrated that dependent models based on voltages substantially impact DG penetration planning. On the other hand, such research works predicted that DG units might be sent and distributed based on peak load demand. According to research work, it has been observed that dynamic load variations models affect the optimal Allocation of DG (ADG). However, the authors did not discuss the effect of non-dispatchable sources on the ADG.
The above reference is included in text and cited as [3] in references.
Cited in Article: The above response is included in Section 1 and references section.
Round 2
Reviewer 1 Report
The referee suggests to publish this carefully revised version of the paper.
Reviewer 2 Report
This version is acceptable.